# Endocytic protein AP180 assembly domain regulates synaptic vesicle size and release in *Caenorhabditis elegans*

Yu Wang[1,2,3☯], Lanxi Wu[2☯¤], Lin Zhang[2], Yongming Dong[2], Aaradhya Pant[2], Yan Liu[2], Jihong Bai [2]*

**1** Fudan University, Shanghai, P. R. China, **2** Basic Sciences Division, Fred Hutchinson Cancer Center, Seattle, Washington, United States of America, **3** School of Life Sciences, Westlake University, Hangzhou, Zhejiang, P. R. China

☯ These authors contributed equally to this work.
¤ Current address: L.W., Scripps Research, La Jolla, California, United States of America
* jbai@fredhutch.org

## Abstract

Neuronal communication relies on neurotransmitter release from synaptic vesicles. The endocytic protein AP180 is critical for efficient vesicle recycling at presynaptic terminals, and its loss impairs neurotransmission, producing reduced release frequency, enlarged synaptic vesicles, and increased quantal amplitude. Yet how AP180 controls vesicle size and whether vesicle size influences release remains unclear. Here, we show that the C-terminal Assembly domain (AD) of AP180 determines vesicle size and thereby regulates release properties in *Caenorhabditis elegans*. An AP180 variant lacking the AD (AP180ΔAD) increases release frequency, contrasting sharply with the reduced transmission in *ap180* null mutants, yet fails to correct the vesicle size or quantal amplitude. These enlarged vesicles evade curvature-dependent inhibition by complexin, a presynaptic regulator of fusion, while remaining dependent on complexin for evoked responses. This selective escape reveals that vesicle size influences release dynamics through curvature-sensing proteins. Replacing the AP180 AD with actin-binding motifs restores normal vesicle size, quantal amplitude, and release frequency, indicating that actin interactions are both necessary and sufficient for AD function. Biochemically, we show that the intrinsically disordered AD forms condensates that enrich actin monomers and nucleate filament assembly, while full-length AP180 couples PIP2-rich membranes to actin filaments. Together, these findings reveal that the AP180 AD regulates synaptic vesicle size through actin binding, establishing vesicle morphology as a key influencer of curvature-dependent release control.

## Introduction

Neurons communicate through the precisely regulated release of neurotransmitters from synaptic vesicles, small membrane-bound organelles that are the fundamental

**Data availability statement:** All relevant data are available in the Supporting information files. This study does not include any original code.

**Funding:** This research was supported by the National Institutes of Health (R01GM127857 to J.B., https://www.nigms.nih.gov). The *Caenorhabditis elegans* Genetics Center is funded by the NIH Office of Research Infrastructure Programs (P40 OD010440, https://orip.nih.gov). The Electron Microscopy Shared Resource at the Fred Hutchinson Cancer Center is supported in part by the NCI Cancer Center Support Grant (P30 CA015704, https://www.cancer.gov). The funders had no role in study design, data collection and analysis, decision to publish, or preparation of the manuscript.

**Competing interests:** The authors have declared that no competing interests exist.

**Abbreviations:** AD, assembly domain; ANTH, AP180 N-terminal Homology; CD, C-terminal intrinsically disordered region; EPSCs, excitatory postsynaptic currents; FRAP, fluorescence recovery after photobleaching; HIPR-1, huntingtin-interacting-protein-related 1; LLPS, liquid–liquid phase separation; MosSCI, Mos1-mediated single-copy insertion; NfM, Neurofilament-M; NGM, nematode growth medium; ROIs, regions of interest; SV, synaptic vesicle.

quanta of synaptic transmission. The concept of quantal neurotransmitter release was first established through electrophysiological recordings at the frog neuromuscular junction [1–3], providing functional evidence for discrete miniature neurotransmission signals. Subsequent ultrastructural analyses revealed a uniform population of synaptic vesicles at presynaptic terminals [4–13], leading to the vesicle hypothesis of synaptic transmission. The demonstration that neurotransmitters are stored within these vesicles [14] further reinforced their central role in chemical synapses, positioning synaptic vesicles as the morphological correlates of quantal neurotransmission [15–17].

Endocytosis plays a crucial role in sustaining neuronal communication by replenishing the vesicle pool and preserving vesicle identity and dimensions necessary for precise neurotransmission [15,18–21]. A key aspect of endocytosis is the shaping of vesicle dimensions, which directly influences neurotransmitter storage capacity and, consequently, quantal amplitude [22,23]. In addition, endocytosis restores essential proteins and lipids to synaptic vesicles, preserving their identity and ensuring reliable transmission. A hallmark of chemical synapses across animal species is the presence of hundreds to thousands of uniformly-sized vesicles, each about 30–50 nm in diameter [18,23–25]. This remarkable uniformity illustrates the conserved ability of endocytosis to reproduce small, consistent vesicle dimensions, suggesting that the mechanisms preserving vesicle size have been evolutionarily maintained.

The endocytic adaptor protein AP180 is central to synaptic vesicle recycling, influencing vesicle morphology and fusogenicity to ensure efficient neurotransmission [25–27]. Loss of AP180 impairs synaptic transmission, disrupts retrieval of synaptic vesicle proteins from the plasma membrane, and leads to abnormally enlarged vesicles in mouse and *Drosophila* neurons [28–31]. Similarly, mutations in the *unc-11* gene, which encodes the *Caenorhabditis elegans* AP180 ortholog, produce comparable defects, including severely reduced locomotion (likely due to impaired synaptic transmission at neuromuscular junctions), accumulation of synaptic vesicle proteins on the plasma membrane, and enlarged synaptic vesicles [32–34]. These findings demonstrate that AP180 has an essential function in preserving synaptic vesicle quality across animal species. Structurally, AP180 proteins feature a modular architecture, characterized by an N-terminal AP180 N-terminal Homology (ANTH) domain [35,36] and a C-terminal intrinsically disordered Assembly domain (AD) [37]. The ANTH domain has been extensively studied for its interaction with phosphatidylinositol 4,5-bisphosphate [35,36] and the v-SNARE protein synaptobrevin/VAMP [38], which mediate membrane binding and vesicle fusion with the plasma membrane [39–41]. By contrast, the intrinsically disordered AD is less well characterized. It is hypothesized to promote membrane curvature during endocytosis through steric pressure arising from its large hydrophobic radius, a property of its disordered nature [42–46]. According to this model, when coupled to the membrane-binding ANTH domain, steric forces generated by the dense packing of multiple AP180 AD copies on membranes induce bending, thereby facilitating vesicle formation.

Here, we investigated the role of the AP180 AD in *C. elegans*, uncovering its importance in maintaining synaptic vesicle morphology and safeguarding synaptic

transmission. We show that loss of the AD results in enlarged synaptic vesicles and increased quantal amplitudes, resembling the defects observed in *ap180* null mutants. Unexpectedly, synapses expressing AP180 lacking the AD exhibit increased, rather than decreased, neurotransmitter release frequency, revealing an inhibitory role of AP180 AD in synaptic transmission. Furthermore, in the absence of AP180 AD, enlarged vesicles evade regulation by complexin, a synaptic protein that senses membrane curvature [70], revealing an unexpected link between vesicle morphology and neurotransmitter release dynamics. Finally, we show that substituting AP180 AD with actin-binding motifs restores both vesicle size and neurotransmitter release frequency, illustrating a functional link between actin cytoskeleton and AP180 AD in synaptic vesicle regulation.

## Results

### Deletion of *unc-11* disrupts synaptic transmission and alters synaptic vesicle dimension in *C. elegans*

To investigate the role of the AP180 ortholog UNC-11 in *C. elegans*, we used CRISPR-Cas9 to generate the *pek217* mutant allele, which removes most of the *unc-11* gene, including part of exon 3, exons 4–7, and introns between these exons (S1A Fig). We then performed electrophysiological recordings at neuromuscular junctions in *pek217* mutants and compared them to another putative null allele, *e47,* previously identified in forward genetic screens. The *e47* allele lacks 210 base pairs spanning portions of exons 1 and 2 and intron 1 [32,34]. As expected, both *pek217* and *e47* mutations lead to significant reductions in synaptic transmission (S1B and S1C Fig). Specifically, the amplitude of evoked excitatory postsynaptic currents (evoked EPSCs) is lower in mutants (*pek217*: 0.20 ± 0.04; *e47*: 0.19 ± 0.04 nA) compared to wild-type N2 worms (3.24 ± 0.20 nA; *p* < 0.001, S1B Fig). Similarly, the frequency of endogenous EPSCs is reduced in both *pek217* and *e47* (*pek217*: 6 ± 1; *e47*: 11 ± 3; versus N2: 53 ± 3 Hz; *p* < 0.001; S1C Fig). In contrast, and consistent with observations in *ap180* knockout mice and flies, the amplitude of endogenous EPSCs is elevated in both *pek217* (31 ± 2 pA) and *e47* (30 ± 2 pA) relative to N2 (22 ± 2 pA; *p* < 0.001; S1C Fig). Collectively, these data show that *unc-11 ap180* is required for normal synaptic transmission in *C. elegans*, mirroring findings in mice and flies. Because *pek217* and *e47* mutants displayed similar phenotypes, we used the *pek217* allele for subsequent experiments, given its more extensive deletion and lower likelihood of carrying non-specific mutations from chemical mutagenesis. Unless otherwise specified, we refer to *pek217* as "*unc-11* mutant" in the text and "*unc-11* mut." in the figures.

### Deleting the AD from UNC-11 increases release frequency yet fails to restore quantal size

To determine the role of the AD, we tested whether full-length UNC-11 (Fig 1A) or a truncated version lacking the AD (UNC-11ΔAD, Fig 1B) could restore synaptic transmission in *unc-11* mutant worms. Both UNC-11 variants were introduced as single-copy transgenes and expressed in neurons under the pan-neuronal promoter *snb-1p*. At the behavioral level, expression of full-length UNC-11 restores locomotion to near wild-type levels (N2: 142 ± 3; full-length (FL): 126 ± 6; versus *unc-11* mutant: 24 ± 1 μm/s; Fig 1C). Interestingly, expressing UNC-11ΔAD in *unc-11* mutants causes an even greater increase in locomotion, surpassing those achieved by full-length UNC-11 (156 ± 3 versus 126 ± 6 μm/s; *p* < 0.001). This suggests that AP180 AD negatively regulates neuromuscular activity, and its removal enhances locomotion.

Full-length UNC-11 fully restores evoked EPSC amplitude (3.1 ± 0.2 nA) and endogenous EPSC frequency (51 ± 6 Hz) to wild-type levels (Fig 1D-1F). Consistent with its effect on locomotion, UNC-11ΔAD expression further increases evoked EPSC amplitude (4.3 ± 0.2 nA; *p* < 0.01) and significantly elevates endogenous EPSC frequency (75 ± 5 Hz; *p* < 0.001) compared to either N2 or *unc-11* mutants rescued with full-length UNC-11 (Fig 1D-1F). These data suggest that the AD of UNC-11 has an inhibitory role in neurotransmitter release, and its removal enhances synaptic transmission, leading to increased neuromuscular activity.

However, despite enhancing evoked EPSC amplitude and endogenous EPSC frequency, expression of UNC-11ΔAD in *unc-11* mutants failed to restore quantal size (UNC-11ΔAD: 37 ± 2 pA, Fig 1E-1G). Endogenous EPSC amplitudes in

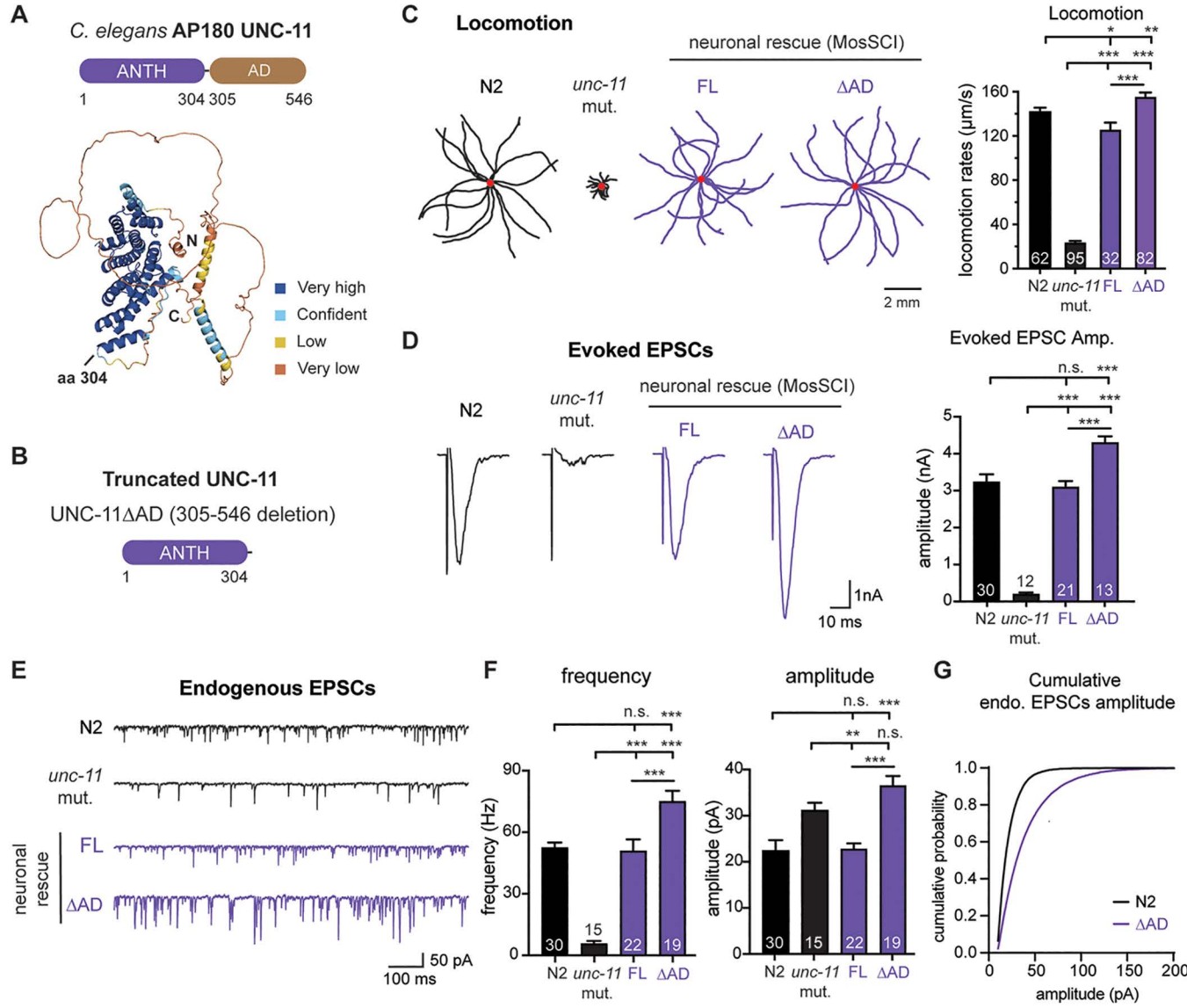

**Fig 1. Removal of the disordered AD region from UNC-11 disrupts synaptic transmission. (A)** Domain organization of UNC-11, showing the N-terminal ANTH domain (residues 1–304) and the C-terminal disordered assembly domain (AD, residues 305–546). *Bottom*: AlphaFold-predicted structure of UNC-11, colored by per-residue model confidence (predicted Local Distance Difference Test score, dark blue: very high; light blue: high; champagne: moderate; brown: low). N and C termini are labeled; "aa 304" marks the ANTH domain boundary. **(B)** Schematic of truncated UNC-11ΔAD, which lacks residues 305–546. **(C–G)** Full-length UNC-11 (FL) and UNC-11ΔAD were expressed in *unc-11(pek217)* mutant worms using the MosSCI system under the pan-neuronal promoter *snb-1p*. "N2" indicates wild-type worms. **(C)** Representative locomotion trajectories (30 sec, 15 worms per genotype; red dots mark start points; scale bar = 2 mm) and quantification of locomotion speed (mean ± SEM). **(D)** Representative traces of evoked EPSCs (*left*) and summary data for evoked EPSC amplitude (*right*) are shown. **(E)** Endogenous EPSC representative traces and **(F)** summary data of the frequency (*left*) and amplitude (*right*) for indicated genotypes are shown. Data are represented as mean ± SEM. Error bars represent SEM. The number of worms per genotype is shown on the bar graphs. One-way ANOVA with Tukey's HSD post hoc test was used. n.s., not significant; *$p < 0.05$; **$p < 0.01$; ***$p < 0.001$. **(G)** Cumulative distribution plot of endogenous EPSC amplitudes for indicated genotypes. The data underlying this figure are provided in S1 Data.

UNC-11ΔAD neurons were not significantly different from those in *unc-11* mutants (*unc-11* mutant: 31 ± 2 pA; not significant; Fig 1F). In contrast, full-length UNC-11 restores endogenous EPSC amplitudes to wild-type levels (FL: 23 ± 1 pA; N2: 22 ± 2 pA; not significant; Fig 1E-1F). These findings indicate that the AD of UNC-11 is essential for maintaining normal

quantal size and plays distinct roles in regulating both quantal size and release frequency, two key properties of synaptic transmission.

Next, we asked whether expressing the AD alone, without the N-terminal ANTH domain, could restore synaptic activity in *unc-11* mutant worms. We introduced a single-copy transgene encoding a truncated UNC-11 (residues 305–546) lacking the ANTH domain (S2A Fig), into *unc-11* mutant worms. However, expressing the UNC-11 AD alone fails to improve locomotion (UNC-11 AD: $25 \pm 1$ μm/s, *unc-11* mutant: $24 \pm 1$ μm/s; not significant; S2B Fig). Furthermore, expression of UNC-11 AD alone generates no significant physiological changes, as it fails to increase evoked EPSC amplitude ($0.22 \pm 0.05$ nA, S2C Fig) or endogenous EPSC frequency ($5.1 \pm 0.5$ Hz; S2D Fig, *left panel*) and has no impact on endogenous EPSC amplitude ($31 \pm 2$ pA; S2D Fig, *right* panel). These findings indicate that while the UNC-11 AP180 AD is necessary for synaptic function, it is not sufficient on its own to support synaptic function.

## UNC-11 ΔAD retains activity in synaptic vesicle protein recycling

Given that UNC-11ΔAD enhances locomotion and synaptic transmission in *unc-11* mutants, we hypothesized that synaptic vesicle endocytosis remains functional at UNC-11ΔAD synapses. To test this, we tracked synaptic vesicle retrieval in the glutamatergic ASH neuron [41,47]. To visualize endocytosis dynamics, we used a pH-sensitive GFP variant, superecliptic pHluorin, inserted into the first lumenal domain of the vesicular glutamate transporter EAT-4 VGLUT, generating a VGLUT-pHluorin reporter (Fig 2A). Under resting conditions, the acidic vesicle lumen quenches VGLUT-pHluorin fluorescence. Upon synaptic vesicle exocytosis, fusion with the plasma membrane exposes VGLUT-pHluorin to the neutral extracellular environment, leading to fluorescence de-quenching and an increase in fluorescence intensity. Subsequent endocytosis and reacidification return the reporter to its quenched state, allowing us to measure vesicle recycling efficiency. As expected, *unc-11 ap180* mutants display significantly slower VGLUT-pHluorin retrieval following stimulation ($\tau = 51 \pm 6$ s) compared to wild-type worms ($19 \pm 3$ s, Fig 2B and 2D), indicating that synaptic vesicle recycling is impaired in neurons lacking UNC-11. Additionally, baseline VGLUT-pHluorin fluorescence is elevated in *unc-11* mutants relative to wild-type worms (Fig 2E), suggesting an accumulation of vesicular proteins at the plasma membrane. These findings are consistent with previous studies showing that AP180 is essential for recycling synaptic vesicle proteins [29,33]. Expression of UNC-11, either full-length or UNC-11ΔAD, fully restores vesicle retrieval rates (FL: $25 \pm 2$ s, and ΔAD: $28 \pm 4$ s; not significant; Fig 2C and 2D) and normalizes basal VGLUT-pHluorin fluorescence to wild-type levels (Fig 2E). These findings show that removing the AD does not impair the ability of UNC-11 AP180 to facilitate endocytic retrieval of synaptic vesicle proteins from the plasma membrane.

## UNC-11 ΔAD synapses accumulate enlarged vesicles

Because UNC-11ΔAD synapses exhibit increased quantal size similar to *unc-11* null mutants, we next asked whether they also display similar ultrastructural defects. Using high-pressure freeze electron microscopy (Fig 3A), we confirmed that *unc-11* mutant synapses contain significantly larger synaptic vesicles ($40.4 \pm 0.2$ nm; Fig 3B *left*) compared to wild-type ($32.1 \pm 0.2$ nm), consistent with prior reports [34]. This enlargement is evident in a right-shifted cumulative distribution of vesicle diameters (Fig 3C) and is accompanied by more frequent endosome-like structures (Fig 3D).

While expression of full-length UNC-11 restores vesicle diameter to wild-type levels ($33.0 \pm 0.3$ nm; Fig 3B), UNC-11ΔAD does not. Instead, synaptic vesicle size in UNC-11ΔAD synapses remains nearly identical to that of *unc-11* mutants ($40.2 \pm 0.3$ nm; Fig 3B *left*) and displays a similarly shifted diameter distribution (Fig 3C). In addition, UNC-11ΔAD synapses continue to exhibit increased presence of endosome-like structures (Fig 3D). These results suggest that the UNC-11 AD is required to maintain normal vesicle size, a function that likely contributes to the elevated quantal size observed in UNC-11ΔAD synapses.

To determine whether this function of the UNC-11 AD is conserved among AP180-family proteins, we replaced UNC-11 AD with the AD from mouse AP180 (mAP180 AD) [29,48] or its homolog CALM (clathrin assembly lymphoid myeloid

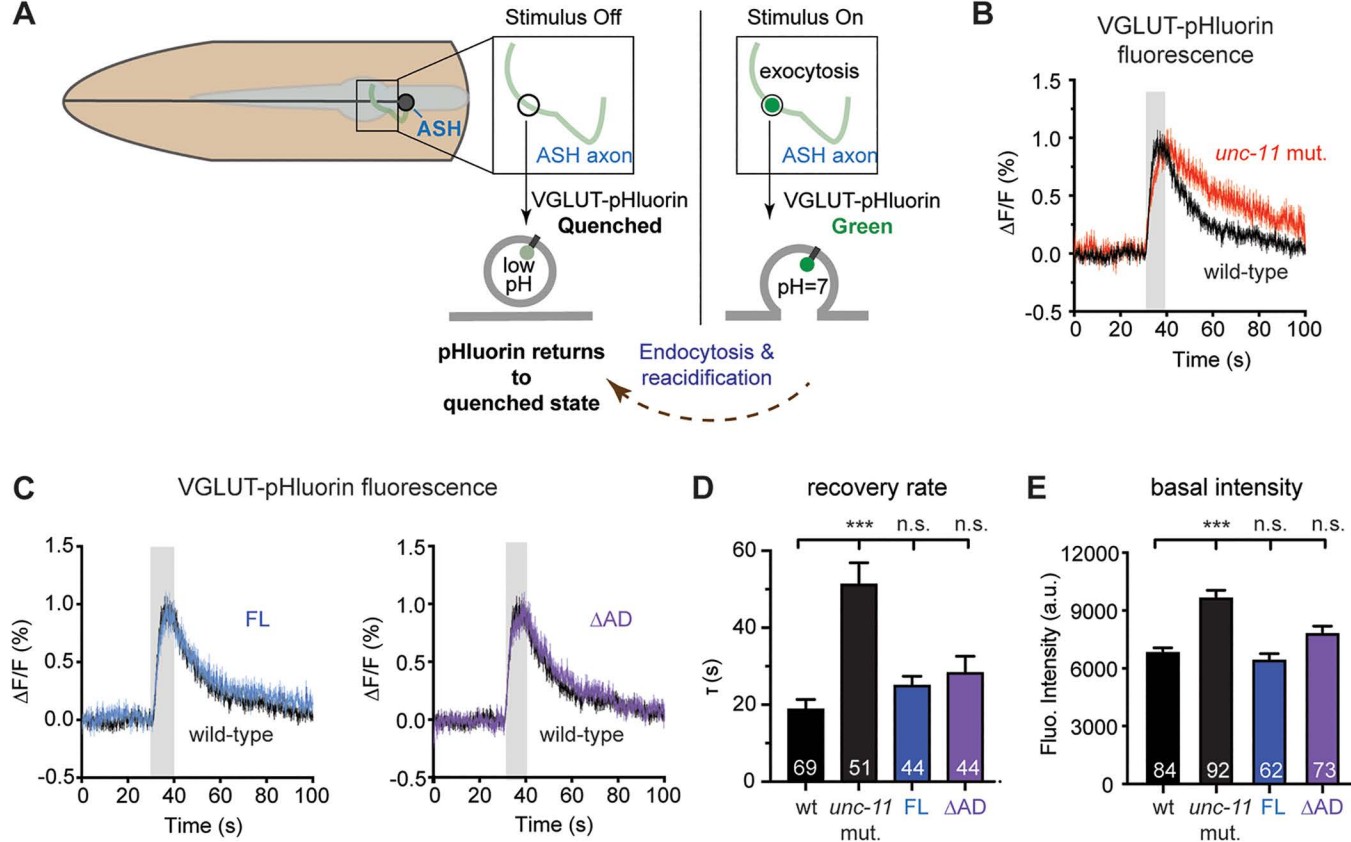

**Fig 2. Internalization of synaptic vesicle proteins is preserved in the absence of UNC-11 AD. (A)** Schematic of VGLUT-pHluorin assay in ASH neurons. A super-ecliptic pHluorin is inserted into the first lumenal domain of EAT-4 (VGLUT) and expressed under the *sar-6p* promoter. At rest, fluorescence is quenched by the acidic vesicle lumen. Upon 0.5 M NaCl stimulation (Stimulus On), VGLUT-pHluorin is exposed to the extracellular space via exocytosis, increasing fluorescence. Following stimulus removal (Stimulus Off), vesicle re-acidification quenches the fluorescence as VGLUT-pHluorin is internalized. **(B)** VGLUT-pHluorin responses over time from wild-type (black) and *unc-11* mutant (red) worms. Grey shading indicates stimulus period; SEM is shown. **(C)** VGLUT-pHluorin responses from wild-type (black) and *unc-11* mutant worms expressing either full-length UNC-11 (FL, *left*, blue) or UNC-11ΔAD (*right*, purple). Summary data of fluorescence recovery rates and basal fluorescence intensity at rest are shown in **(D)** and **(E)**, respectively. Data are shown as mean±SEM. *n* values are indicated on bar graphs. *** *p* < 0.001 vs. wild type (one-way ANOVA with Tukey's HSD post hoc test); n.s., not significant. The data underlying this figure are provided in S2 Data.

leukemia, [49,50]), and expressed these chimeric constructs as single-copy transgenes (S3A Fig). Both mAP180 AD and mCALM AD significantly reduce synaptic vesicle diameter, from about 40 nm in *unc-11* mutants to 35.4±0.5 nm and 35.6±0.3 nm, respectively (*p* < 0.001; S3B Fig). These findings indicate the conserved role of C-terminal disordered AD regions of AP180 in controlling synaptic vesicle size.

## Enlarged synaptic vesicles at UNC-11ΔAD synapses evade complexin-mediated suppression of endogenous EPSCs

Our observation that AP180ΔAD synapses exhibit elevated endogenous EPSC frequency (Fig 1E) without an accompanying increase in synaptic vesicle abundance (Fig 3B *right*) suggests that these vesicles bypass normal inhibitory mechanisms that limit vesicle fusion. One such mechanism involves complexin (CPX-1), a presynaptic protein that suppresses endogenous EPSCs in *C. elegans* through SNARE binding and curvature sensing [51–53], yet promotes evoked release through curvature-independent, SNARE-dependent mechanisms [52,53].

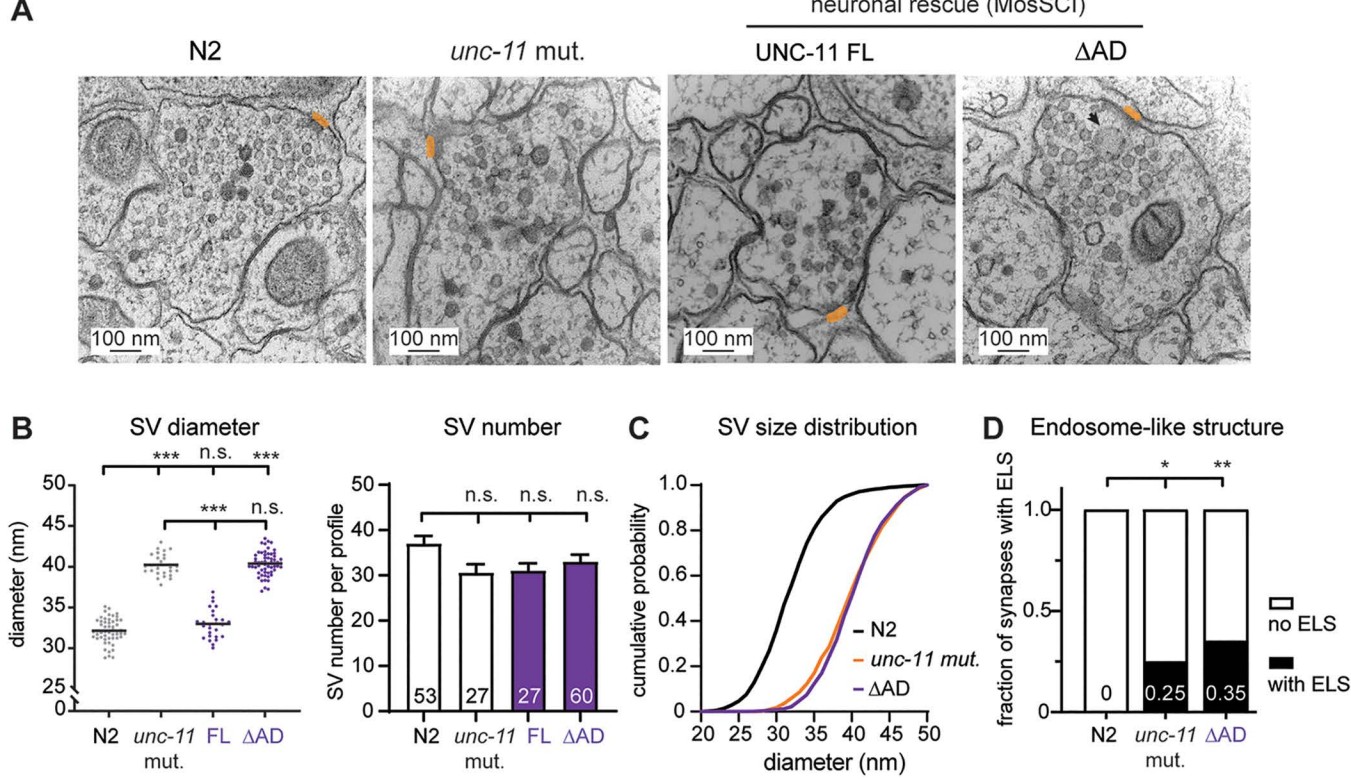

**Fig 3. UNC-11 ΔAD synapses exhibit enlarged synaptic vesicles and abnormal endosome-like structures. (A)** Representative transmission electron micrographs of synapses from wild-type N2, *unc-11* mutants, Full-length UNC-11-rescued, and UNC-11ΔAD-rescued *unc-11* mutants. Dense projections are highlighted in orange. The arrow marks an endosome-like structure (ELS, >100 nm in diameter). Scale bar, 100 nm. **(B)** *Left*: Scatter plot showing the mean synaptic vesicle (SV) diameter for individual synaptic profiles. Mean values are indicated by the horizontal lines on the graph. Each data point represents a single profile. *Right*: Bar graph summarizing the number of vesicles per synaptic profile, with the numbers of synaptic profiles analyzed indicated on the bars. Data are shown as mean ± SEM. **(C)** Cumulative distribution plot of vesicle diameter at synapses for indicated genotypes. **(D)** The fraction of synapses containing ELS is shown in bar graphs for the indicated genotypes. Continuous variables were analyzed by one-way ANOVA with Tukey's HSD post hoc test; categorical variables by Fisher's exact test. n.s., not significant; * $p < 0.05$; ** $p < 0.01$; *** $p < 0.001$. The data underlying this figure are provided in S3 Data.

To test whether CPX-1–mediated inhibition on endogenous EPSC is impaired at AP180/UNC-11ΔAD synapses, we examined the effect of removing *cpx-1*. Consistent with previous findings [53,54], *cpx-1* mutants show significantly elevated endogenous EPSC frequency compared to wild-type animals, with no change in amplitude (Fig 4A). However, at UNC-11ΔAD synapses, *cpx-1* deletion has no effect on either frequency or amplitude at 1 and 0.25 mM external Ca$^{2+}$ (Figs 4A and S4A), suggesting that the inhibitory function of CPX-1 is already compromised when vesicles are enlarged. In contrast, evoked EPSC amplitudes are significantly reduced in *unc-11ΔAD; cpx-1* double mutants to levels comparable to *cpx-1* null mutants (Fig 4B), indicating that enlarged vesicles remain dependent on the curvature-independent facilitation of evoked release by CPX-1.

CPX-1 inhibition of endogenous EPSC requires both SNARE binding and curvature-dependent membrane interactions [51–53]. Previous work showed that the curvature requirement can be bypassed by tethering CPX-1 to the synaptic vesicle-associated protein RAB-3 (Fig 4C) [51]. To test whether anchoring CPX-1 to synaptic vesicles could restore its inhibitory function at UNC-11ΔAD synapses, we expressed a CPX-1::RAB-3 chimera (Fig 4D). This construct significantly reduces endogenous EPSC frequency to 24 ± 6 Hz at UNC-11ΔAD synapses without affecting EPSC amplitude (Fig 4E).

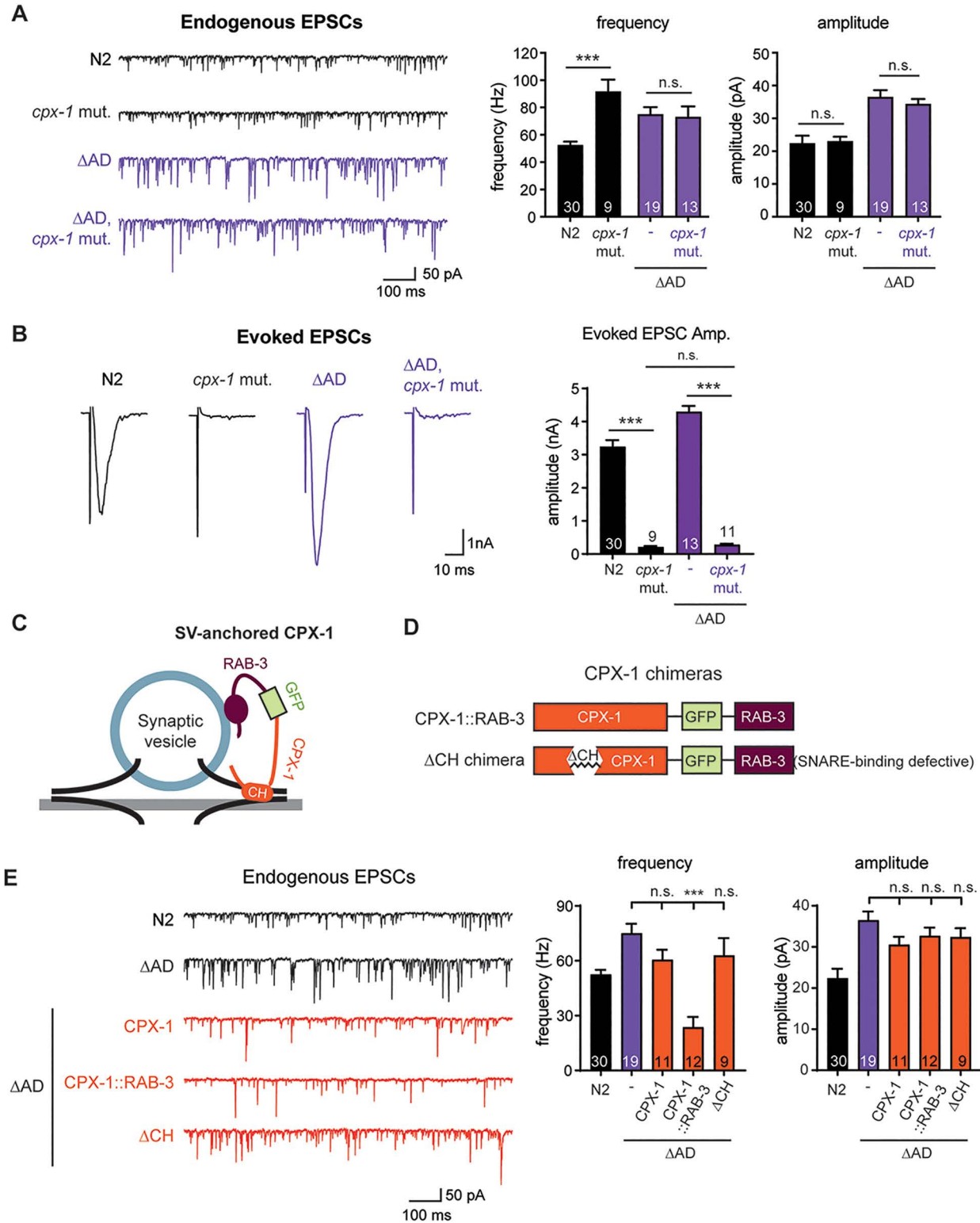

**Fig 4. Complexin inhibition of vesicle fusion is impaired at UNC-11ΔAD synapses but remains essential for evoked release. (A)** Representative traces (*left*) and summary data (*middle, right*) of endogenous EPSC frequency and amplitude for the indicated genotypes. "*cpx-1* mut." refers to worms carrying the *cpx-1(ok1552)* deletion allele. "ΔAD" indicates *unc-11* mutant worms expressing UNC-11ΔAD in neurons, and "ΔAD, *cpx-1* mut." indicates

ΔAD worms also carrying the *cpx-1* mutation. Data are presented as mean±SEM. Unpaired Student *t* test; *** *p* < 0.001; n.s., not significant. **(B)** Representative traces of evoked EPSCs (*left*) and summary data for evoked EPSC amplitude (*right*) are shown. **(C)** Schematic of synaptic vesicle-anchored CPX-1 (CPX-1::RAB-3) design. **(D)** Schematic of CPX-1::RAB-3 variants: full-length CPX-1 (*top*) and a ΔCH chimera lacking the SNARE-binding central helix (*bottom*). Chimeras were expressed in *unc-11* mutant worms via single-copy transgenes under the pan-neuronal *snb-1p* promoter. **(E)** Representative traces (*left*) and summary data (*middle, right*) of endogenous EPSC frequency and amplitude for indicated genotypes. "ΔAD, CPX-1", "ΔAD, CPX-1::RAB-3", and "ΔAD, ΔCH" indicate ΔAD worms expressing wild-type CPX-1, the CPX-1::RAB-3 chimera, or the ΔCH variant of the CPX-1::RAB-3 chimera, respectively. Data are shown as mean±SEM. One-way ANOVA with Tukey's HSD post hoc test; *** *p* < 0.001; n.s., not significant. The data underlying this figure are provided in S4 Data.

In contrast, overexpression of wild-type CPX-1 failed to reduce the elevated mini frequency (Fig 4E). These results indicate that directly targeting CPX-1 to synaptic vesicles can restore its inhibitory function, supporting the idea that enlarged vesicle size at UNC-11ΔAD synapses compromises CPX-1 recruitment and thus impairs normal synaptic regulation.

To test whether CPX-1 still requires SNARE binding for its inhibitory function in this context, we expressed a CPX-1::RAB-3 variant lacking the central helix (ΔCH; Fig 4D), which disrupts SNARE interactions. This mutant fails to suppress EPSC frequency at AP180ΔAD synapses (Fig 4E), demonstrating that SNARE binding is still essential and cannot be bypassed by RAB-3 tethering.

Collectively, these findings suggest that enlarged synaptic vesicles at UNC-11ΔAD synapses escape CPX-1 inhibition of endogenous EPSC because their altered size disrupts curvature-dependent CPX-1 recruitment, while evoked release continues to rely on curvature-independent, SNARE-dependent facilitation.

### Synaptic vesicles remain enlarged despite tethering disordered motifs to UNC-11ΔAD to enhance steric pressure

To investigate how the AP180 AD controls synaptic vesicle size, we considered a recent model proposing that the AD promotes membrane curvature through steric pressure generated by its large hydrophobic radius [42–46]. In this model, when coupled to the membrane-binding ANTH domain, steric forces generated from densely-packed AP180 AD copies induce membrane bending, facilitating vesicle formation. Supporting this idea, biophysical analyses show that chimeric AP180 proteins, in which the AD was replaced by the C-terminal intrinsically disordered region (CD) of Neurofilament-M (NfM), a protein lacking membrane affinity, retained curvature-inducing activity comparable to that of the native AP180 AD in vitro [45]. These findings suggest a model in which the AD regulates vesicle size through its disordered nature rather than through specific interactions with proteins or membranes.

To test whether increasing the hydrophobic radius of disordered motifs could restore synaptic vesicle size in vivo, we generated a chimeric UNC-11 protein by fusing the NfM CD to UNC-11ΔAD (Fig 5A), a strategy previously used in vitro [42,45]. Electron microscopy analysis revealed that synaptic vesicles at UNC-11ΔAD::NfM-CD synapses remain enlarged (38.8±0.4 nm; Fig 5B and 5C) compared to wild-type (32.1±0.2 nm; *p* < 0.001), though they are slightly but significantly reduced relative to those in *unc-11* mutants (40.2±0.2 nm; *p* < 0.05) or UNC-11ΔAD synapses (40.4±0.2 nm; *p* < 0.001). Additionally, endosome-like structures appear less frequently at UNC-11ΔAD::NfM-CD synapses compared to UNC-11ΔAD synapses, though this difference was not statistically significant (Fig 5D, *p* = 0.14). These results suggest that increasing the hydrophobic radius by tethering disordered motifs provides a modest improvement in synaptic vesicle size and membrane organization at synapses but is not sufficient to fully account for the role of UNC-11/AP180 in synaptic vesicle size control.

### Tethering HIPR-1 and Epsin1 C-terminal domains to UNC-11ΔAD improve vesicle size distribution.

To investigate the mechanisms that determine synaptic vesicle size, we considered an alternative model in which the AP180 AD regulates vesicle dimensions through protein interactions shared among other endocytic proteins. This hypothesis was based on three key observations. First, several endocytic proteins, including AP180,

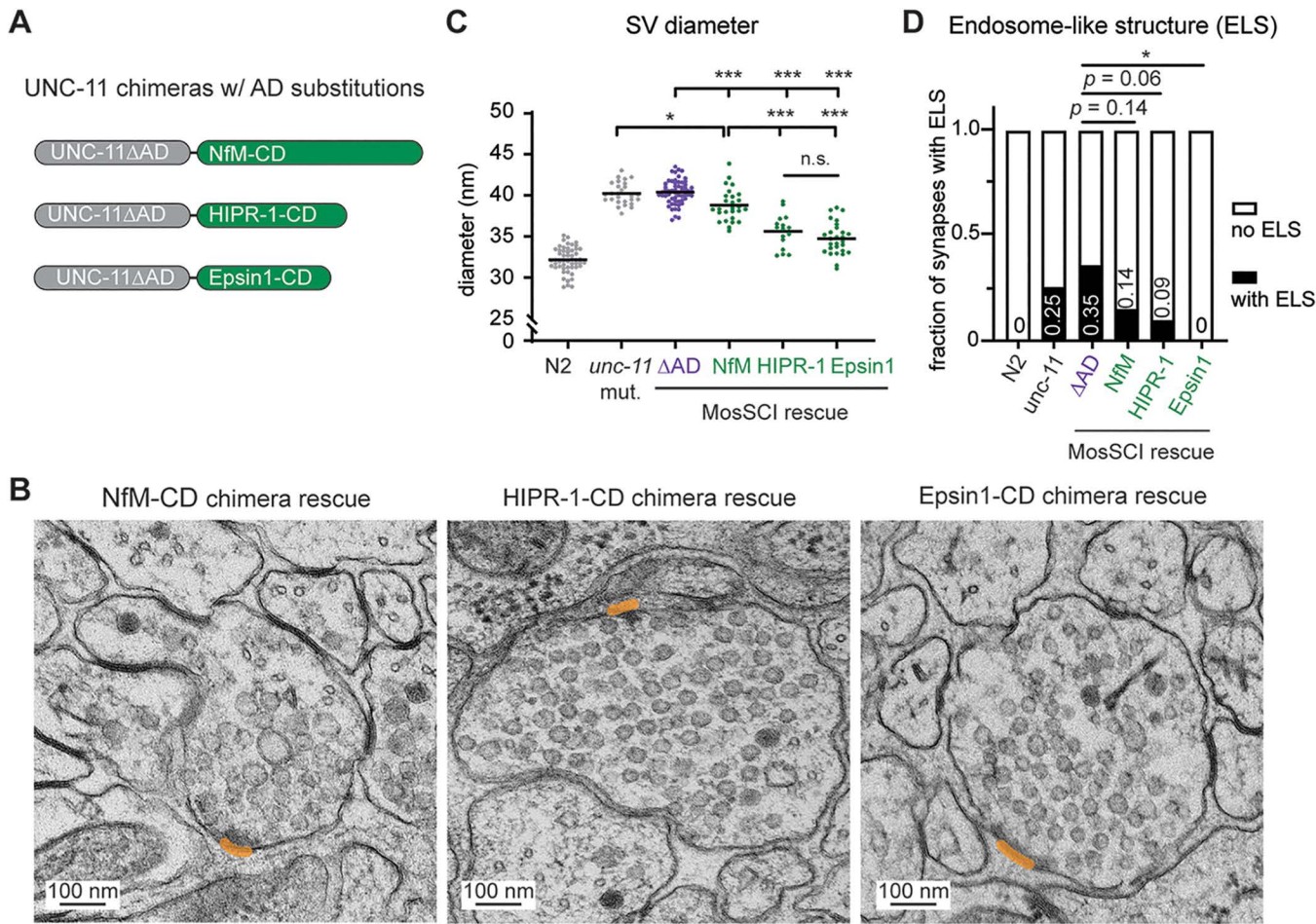

**Fig 5. C-terminal domains from HIPR-1 and Epsin1 substituted UNC-11 AD for synaptic vesicle (SV) size regulation. (A)** Schematics of UNC-11 chimeras in which the UNC-11 ANTH domain (residues 1–304) is fused to the C-terminal intrinsically disordered region of mouse Neurofilament-M (NfM-CD, residues 411–848), the C-terminal domain of *Caenorhabditis elegans* HIPR-1 (HIPR1-CD, residues 636–928), or the C-terminal domain of rat Epsin1 (Epsin1-CD, residues 316–575). Chimeras were expressed in *unc-11* mutant worms via single-copy transgenes driven by the pan-neuronal *snb-1p* promoter. "NfM," "HIPR-1," and "Epsin1" refer to *unc-11* mutants expressing each chimera. **(B–D)** EM analysis of synaptic vesicle dimensions at *unc-11* mutant synapses expressing the chimeric UNC-11 proteins. **(B)** Representative images of transmission electron micrographs for each genotype. Dense projections are labeled in orange. **(C)** Scatter plot showing the average synaptic vesicle diameter per synaptic profile. Mean values are indicated by the horizontal lines on the graph. Each data point represents one synaptic profile. **(D)** The fraction of synapses containing ELS is shown in bar graphs for the indicated genotypes. Vesicle diameter data analyzed using one-way ANOVA followed by Tukey's HSD post hoc test; percentages of synapses with ELS analyzed using Fisher's exact test. * $p < 0.05$; *** $p < 0.001$. The data underlying this figure are provided in S5 Data.

huntingtin-interacting-protein-related 1 (HIPR-1), and Epsin1, share a similar domain organization, consisting of an N-terminal membrane-binding ANTH/ENTH domain [35,55,56] and a large C-terminal region [37,57]. Second, these proteins interact [58–60] and function cooperatively to facilitate endocytosis [29,58,61,62]. Third, deletion of individual proteins within this group results in impaired endocytic vesicles in cells [58,63,64]. Supporting this model, we find that deletion of *hipr-1*, the *C. elegans* ortholog of yeast Sla2 and human HIP1/HIP1R, results in increased quantal size, elevated endogenous EPSC frequency, and enlarged synaptic vesicles (S5A–S5D Fig). Endosome-like structures were occasionally observed but did not reach statistical significance (S5E Fig). These defects at *hipr-1* mutant synapses closely resemble those observed at UNC-11ΔAD synapses, suggesting that HIPR-1 and UNC-11 converge on a shared mechanism of vesicle size regulation.

These findings prompted us to test whether the C-terminal regions of HIPR-1 and Epsin1 can functionally substitute for the AP180 AD in regulating vesicle size in vivo. To do so, we generated UNC-11ΔAD chimeras by fusing UNC-11ΔAD to HIPR-1-CD or Epsin1-CD (Fig 5A). Expression of these chimeric proteins in *unc-11* mutants significantly reduces synaptic vesicle size (HIPR-1-CD: 35.6 ± 0.5 nm, and Epsin1-CD: 34.7 ± 0.4 nm, respectively) compared to those at UNC-11ΔAD synapses or UNC-11ΔAD::NfM-CD synapses (Fig 5B-5C). Also, the occurrence of endosome-like structures is lower at UNC-11ΔAD::HIPR-1-CD and UNC-11ΔAD::Epsin1-CD synapses compared to UNC-11ΔAD synapses (Fig 5D). These findings suggest that the C-terminal regions of HIPR-1 and Epsin1 can effectively substitute for the UNC-11 AP180 AD in regulating synaptic vesicle size, supporting the idea that these endocytic proteins contribute to vesicle size regulation through a shared mechanism.

## Actin interactions are necessary and sufficient to restore synaptic vesicle size

Unlike the UNC-11 AP180 AD, which lacks an ordered three-dimensional structure, the HIPR-1 C-terminal domain contains structured regions, including the highly conserved THATCH (talin/HIP1R/Sla2p actin tethering C-terminal homology) domain, which binds actin (S5A Fig) [65–67]. To determine whether actin binding is involved in vesicle size regulation, we deleted the THATCH domain (ΔTHATCH) from the UNC-11ΔAD::HIPR-1-CD chimera (Fig 6A *upper*). This deletion

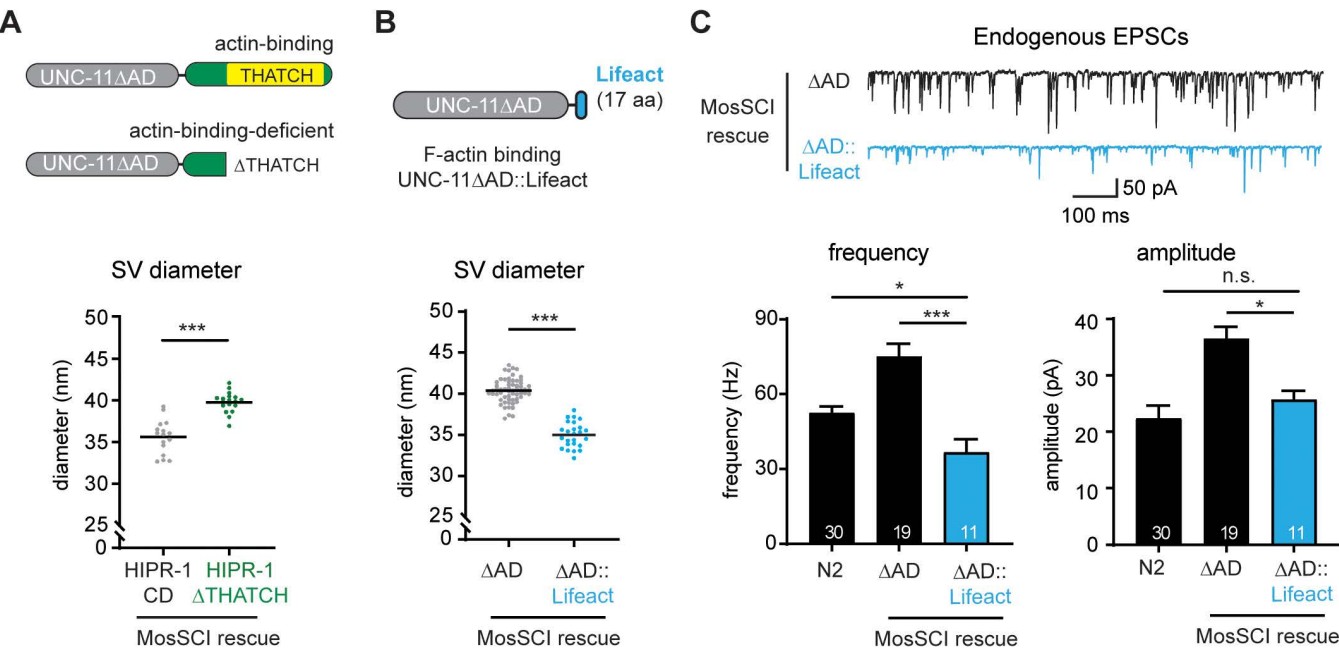

**Fig 6. Deletion of the actin-binding motif from HIPR-1-CD impairs its ability to substitute for UNC-11 AD in regulating synaptic vesicle size.**
**(A)** (*Upper*) Schematics of UNC-11ΔAD::HIPR-1-CD chimeras. The actin-binding THATCH motif is highlighted (yellow). The ΔTHATCH variant lacks the C-terminal THATCH domain (deletion of 207 amino acids). Chimeras were expressed in *unc-11* mutant worms using a single-copy transgene under the pan-neuronal *snb-1p* promoter. "ΔTHATCH" indicates *unc-11* mutant worms expressing the ΔTHATCH variant. (*Lower*) Scatter plot of average vesicle diameter per synaptic profile in UNC-11ΔAD::HIPR-1-CD and ΔTHATCH-rescued unc-11 synapses. Each dot represents a single synaptic profile; horizontal bars indicate mean. **(B)** (*Upper*) Schematic of a UNC-11 variant with a 17-aa F-actin binding motif, Lifeact (blue), fused to the C-terminus of UNC-11ΔAD (ΔAD::Lifeact). The Lifeact amino acid sequence is "MGVADLIKKFESISKEE". This construct is expressed in *unc-11* mutant worms via a single-copy transgene driven by the pan-neuronal *snb-1p* promoter. (*Lower*) Scatter plot of average vesicle diameter per profile in ΔAD- and ΔAD:: Lifeact-rescued unc-11 synapses. Each dot represents one profile; horizontal bars show means. **(C)** Excitatory postsynaptic currents (EPSCs) recorded at neuromuscular junctions. Representative traces (*upper*) and summary data (*lower*) of endogenous EPSC frequency (*left*) and amplitude (*right*) are shown. Data are presented as mean ± SEM. The number of worms analyzed is indicated in the bar graphs. One-way ANOVA followed by Tukey's HSD post hoc test; * *p* < 0.05; *** *p* < 0.001. The data underlying this figure are provided in S6 Data.

disrupts chimera's ability to restore normal vesicle size, resulting in enlarged synaptic vesicles (39.8 ± 0.3 nm, Fig 6A *lower*). These findings suggest that actin interactions are required for generating synaptic vesicles of normal size.

Next, we tested whether enhancing actin binding, without significantly altering the hydrophobic radius, could enable UNC-11ΔAD to restore vesicle size. To do this, we fused Lifeact, a 17-amino acid F-actin-binding peptide, to the C-terminus of UNC-11ΔAD (Fig 6B *upper*). Strikingly, adding Lifeact to UNC-11ΔAD significantly restores synaptic vesicle size to 35.0 ± 0.3 nm (Fig 6B *lower*). Functionally, expressing the UNC-11ΔAD::Lifeact chimera in *unc-11* mutant worms rescues endogenous EPSC frequency (37 ± 5 Hz) and amplitude (26 ± 2 pA) (Fig 6C). These results demonstrate that strengthening actin interactions enables AP180ΔAD to regulate synaptic vesicle size, highlighting actin binding as both essential and sufficient for AP180-mediated vesicle size control.

To test whether the UNC-11 AD domain can functionally substitute for the HIPR-1 THATCH domain in vivo, we expressed a chimeric HIPR-1 protein in which the THATCH domain was replaced by UNC-11 AD (HIPR-1ΔTHATCH::AD) in *hipr-1* mutant neurons (S5F Fig). This chimera restored endogenous EPSC frequency (56 ± 5 Hz) and amplitude (23.5 ± 1 pA) to wild-type levels (S5G Fig), indicating that the UNC-11 AD domain can replace the actin-binding HIPR-1 THATCH domain in vivo, supporting a model in which both domains share a common mechanistic role in linking endocytosis to the actin cytoskeleton.

## UNC-11 condensates couple actin assembly to PIP2-rich membranes

Our genetic studies indicate that actin binding is essential for AP180-mediated vesicle size control and that the UNC-11 AD domain can functionally replace the HIPR-1 THATCH domain. To determine whether UNC-11 directly interacts with actin, we performed biochemical experiments using recombinant proteins. We expressed and purified full-length UNC-11, the isolated AD region, and the ANTH domain lacking the AD (ANTHΔAD). Both full-length UNC-11 and the isolated AD region spontaneously formed spherical condensates in vitro, exhibiting hallmark features of liquid–liquid phase separation (LLPS): salt-sensitive droplet formation, fluorescence recovery after photobleaching, and fusion upon contact (S6A–S6C Fig). In contrast, the folded ANTH domain alone formed only amorphous aggregates (S6D Fig). These results demonstrate that the intrinsically disordered AD region is necessary and sufficient for LLPS, consistent with findings from yeast AP180 homologs [68].

To test whether UNC-11 condensates interact directly with actin, we introduced fluorescent monomeric G-actin into the system. G-actin rapidly partitioned into both full-length UNC-11 and AD droplets, with faster enrichment in full-length UNC-11 condensates (S7A and S7B Fig). Within minutes, F-actin filaments emerged from UNC-11 droplets, as visualized by phalloidin staining (Fig 7A), demonstrating that UNC-11 condensates nucleate and accelerate actin polymerization. Furthermore, both full-length UNC-11 and AD droplets co-localized with assembled actin filaments (S7B Fig), indicating direct binding to F-actin.

We next asked whether full-length UNC-11 can simultaneously engage membranes and actin to bridge these two cellular compartments. Full-length UNC-11 condensates, but not AD droplets lacking the membrane-binding ANTH domain, formed tri-partite complexes at the interface of PIP2-containing liposomes and F-actin (Figs 7B, 7C, and S8A). This interaction was specific for PIP2, as liposomes composed only of phosphatidylcholine showed minimal association with UNC-11 condensates (Figs 7C and S8B).

Together, these biochemical results show that UNC-11 interacts with actin. UNC-11 forms LLPS condensates that enrich monomeric actin, accelerate actin polymerization, bind assembled actin filaments, and tether F-actin to PIP2-rich membranes. These findings elucidate a molecular mechanism by which UNC-11 links the actin cytoskeleton to endocytic membranes through complementary functions of its AD and ANTH domains.

## Discussion

Neuronal communication depends on both the release probability of synaptic vesicles and their capacity to store neurotransmitters. Here, we identify a novel role for the AP180 AD in regulating vesicle size through actin interactions, which in turn influences release frequency and synaptic transmission strength. We show that removing the AD produces

PLOS Biology

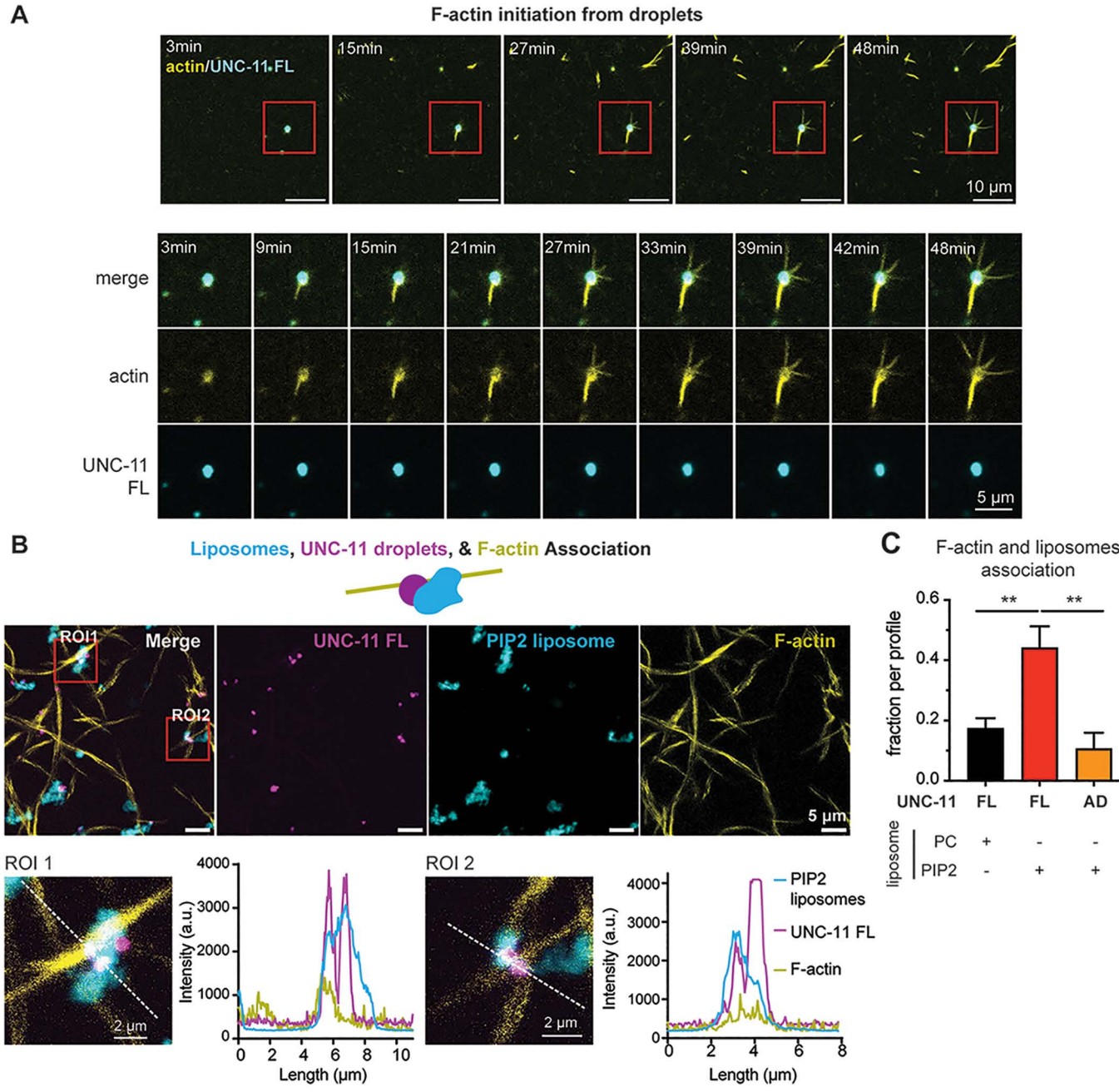

**Fig 7. UNC-11 condensates support actin polymerization and bridge F-actin to PIP₂-containing membranes. (A)** UNC-11 condensates promote the growth of F-actin. Full-length UNC-11 (UNC-11 FL) was labeled with Pacific Blue (cyan), and F-actin was visualized using Alexa Fluor 488–phalloidin (yellow). G-actin (2 μM) and phalloidin (0.2 μM) were added to samples containing UNC-11 droplets, and fluorescence images were collected every 3 min for 48 min. Low-magnification images are shown (*top*), with red rectangles marking regions shown at higher magnification below. Scale bar, 5 μm. Images were acquired on an Olympus FV1000 laser-scanning confocal microscope using a 60×, 1.4 NA objective with 5× zoom. **(B)** UNC-11 condensates connect F-actin to liposomes containing PIP2. A schematic model is shown (*top*). Representative fluorescence images with two regions of interest (ROIs, red boxes) are shown (*middle*). UNC-11 FL is shown in magenta (Pacific Blue), F-actin in yellow (Alexa Fluor 488–phalloidin), and liposomes in cyan (500 μM total lipids; 2% PI(4,5)P2, 25% PS, 71% PC, and 2% Rhodamine-PE; 50 nm diameter). Zoomed-in views and fluorescence intensity line scans of the ROIs are shown (*bottom*). Images were acquired on an Olympus FV1000 confocal system using a 60×, 1.4 NA objective with 5× zoom. **(C)** Quantification of UNC-11 condensates associated with both actin filaments and membranes. The fraction of dual association was calculated from total condensates per Z-stack. Data represent three independent experiments. $p < 0.01$, one-way ANOVA with Dunnett's multiple comparisons. The data underlying this figure are provided in S7 Data.

enlarged vesicles that release more frequently, thereby altering both synaptic transmission rates and quantal strength. Vesicle dimensions at UNC-11ΔAD synapses closely resemble those observed in *ap180* null mutants, indicating that the AD is required to produce normally sized vesicles. We find that enlarged vesicles at AP180ΔAD synapses escape curvature-dependent inhibition by complexin, leading to elevated basal release, as reflected by increased endogenous EPSC frequency, while remaining dependent on complexin for evoked release. Importantly, substituting the AP180 AD with actin-binding motifs rescues both vesicle size and release frequency. Biochemically, we demonstrate that the intrinsically disordered AD forms condensates that directly interact with actin, enriching actin monomers, nucleating polymerization, and binding filaments, while full-length AP180 bridges PIP2-rich membranes to the actin cytoskeleton. Taken together, our findings demonstrate that the AP180 AD safeguards synaptic transmission fidelity by ensuring vesicle size through actin engagement, enabling precise curvature-dependent control of basal vesicle fusion.

## Vesicle size as a key modulator of synaptic transmission

Synaptic vesicles represent the fundamental units of neurotransmitter release. Across diverse animal species, presynaptic terminals maintain uniformly small vesicles [18,23–25], suggesting an evolutionarily conserved mechanism for vesicle size regulation. However, despite this well-documented morphological feature, the functional consequences of vesicle size uniformity are unclear. Our data reveal that loss of the AP180 AD leads to vesicle enlargement, subsequently increasing quantal amplitude and basal release frequency without altering vesicle number. Increased release frequency at UNC-11ΔAD synapses, despite normal vesicle abundance, indicates that vesicle enlargement itself can influence fusion probability. This size-dependent modulation of basal release is not observed in the *unc-11*/*ap180 null* mutant. Although vesicles are similarly enlarged in this background, loss of the AP180 ANTH domain disrupts SNB-1/VAMP2 sorting and recycling to synaptic vesicles. The resulting depletion of vesicle-associated SNAREs substantially decreases fusion competence, thereby masking the increased fusion probability expected from vesicle enlargement.

Our results identify a mechanism by which vesicle size influences neurotransmitter release through its impact on the curvature-sensitive presynaptic regulator complexin. Complexin targeting to synaptic vesicles is curvature-dependent, with a preference for highly curved membranes [69,70]. At UNC-11ΔAD synapses, where vesicles are enlarged, complexin recruitment is likely reduced, weakening its inhibitory function and leading to higher release frequency. Supporting this model, artificially tethering complexin to vesicles via RAB-3 bypasses the curvature requirement and restores its clamping function at UNC-11ΔAD synapses. This suggests that maintaining small vesicle size is essential for complexin-mediated suppression of vesicle fusion, revealing an additional layer of presynaptic regulation dependent on vesicle dimensions. Together, our findings show that synaptic vesicle size is not merely a structural feature but an important influencer of neurotransmitter release dynamics, directly linking vesicle morphology to presynaptic regulatory mechanisms. It is plausible that increased fusogenicity results not solely from altered vesicle morphology, but also from changes in vesicle protein networks. For example, AP180 participates in VAMP2 sorting during endocytosis [28,29,33,34,38,71,72]. However, several lines of evidence suggest that morphological control is a primary driver of the release phenotype. In support of this, loss of HIPR-1, which unlike AP180 is not an adaptor for vesicle protein sorting, produces similar increases in vesicle size and release frequency. Both the enlarged vesicle phenotype and elevated release frequency are rescued by restoring actin interactions, indicating that mechanisms beyond vesicle protein sorting contribute substantially to enhanced release probability. Additionally, the selective loss of curvature-dependent complexin function at UNC-11ΔAD synapses, while SNARE-dependent functions remain intact, is more consistent with a geometric mismatch than with altered protein composition. While the convergence of genetic evidence strongly supports vesicle morphology as a key determinant of release properties, fully establishing causality without confounding influences from vesicle composition would likely require biochemical isolation of synaptic vesicles from ap180ΔAD or *hipr-1* mutant synapses followed by quantitative proteomic analysis.

## Contributions of steric pressure and actin interactions to synaptic vesicle size

Having established that vesicle size influences release through curvature-dependent mechanisms, we next examined how the AP180 AD controls vesicle dimensions. We find that AP180 AD controls synaptic vesicle dimensions through an integrated mechanism involving steric pressure and actin interactions. The AD is intrinsically disordered and has been proposed to promote vesicle curvature by generating entropic repulsion—steric pressure that resists expansion and favors small vesicle formation [43,44,46]. Consistent with this model, deletion of the AP180 AD leads to vesicle enlargement, similar to the phenotype observed in *ap180* null mutants. However, steric effects alone do not fully account for size control. Substituting the AP180 AD with another intrinsically disordered domain—the C-terminal region of NfM—only partially rescues vesicle morphology. This suggests that additional molecular inputs are required to achieve precise vesicle sizing.

Actin network components are important constituents of the endocytic network [73–80], and we show that actin dynamics provide a robust and complementary mechanism for maintaining vesicle size. Introducing actin-binding motifs into the UNC-11ΔAD construct effectively normalizes vesicle dimensions, pointing to a functional role for actin-mediated mechanical forces during endocytosis. Similarly, replacing the AP180 AD with the C-terminal region of HIPR-1, which includes actin-binding THATCH domains, also rescues vesicle morphology. These results underscore the importance of actin interactions in shaping vesicles.

We propose that AP180, HIPR-1, and Epsin 1 function as an integrated endocytic module that couples membrane deformation to actin polymerization, generating the forces necessary for controlled vesicle formation. The actin-binding activity of the AP180 AD positions AP180 as a molecular organizer that coordinates membrane recognition by the ANTH domain with actin assembly at endocytic sites. Loss of actin-binding capacity in one component may trigger compensatory expansion of endocytic membranes to increase actin contact sites, thereby maintaining force balance and bending efficiency. This could explain why loss of either AP180 AD or HIPR-1 produces similar vesicle enlargement phenotypes.

## AP180 condensates interact with actin

Our genetic and biochemical studies demonstrate that the AP180 AD binds actin. Two lines of genetic evidence support actin binding as the key function of the AD. First, fusing the actin-binding Lifeact peptide to UNC-11ΔAD restores both vesicle size and synaptic function, demonstrating that direct interaction with actin is sufficient to rescue AP180 activity. Second, the UNC-11 AD can functionally substitute for the actin-binding THATCH domain of HIPR-1 in vivo, restoring normal synaptic transmission. These experiments show that actin engagement is both necessary and sufficient for AD function in vesicle size control.

Biochemical results revealed that both the isolated UNC-11 AD and full-length UNC-11 form liquid-liquid phase-separated condensates in vitro, consistent with the intrinsically disordered nature of the AD. These condensates enrich monomeric G-actin, nucleate actin polymerization, and bind F-actin filaments. Full-length UNC-11 condensates also assemble tri-partite complexes at the interface of PIP2-containing membranes and F-actin, bridging these compartments through the dual activities of the ANTH and AD domains. This bifunctional architecture allows AP180 to simultaneously recognize PIP2-rich endocytic membranes and recruit actin assembly machinery to these sites.

Although the AD forms condensates that organize actin assembly in vitro, our genetic results indicate that phase separation itself is not required for function. The Lifeact peptide, which binds actin but does not form condensates, fully rescues the UNC-11ΔAD phenotype. This suggests that actin binding is the core functional requirement, whereas condensate formation may enhance the local concentration and organization of actin assembly factors at endocytic sites but is not obligatory. The intrinsically disordered nature of the AD may have evolved to facilitate both actin binding and the formation of phase-separated domains that spatially organize the endocytic machinery.

In summary, our findings support a dual mechanism in which steric constraints from disordered domains and mechanical tension from the actin network act together to ensure precise control of synaptic vesicle size. The essential role of

actin binding, demonstrated through both genetic and biochemical investigations, reveals that AP180 functions not only through passive steric effects but also through active organization of the actin cytoskeleton. This integrated strategy likely provides the robustness required to maintain vesicle uniformity and quantal precision in neurotransmission. Moreover, the finding that vesicle size influences release probability through curvature-sensing proteins like complexin identifies a previously unrecognized form of presynaptic regulation. Given the conservation of synaptic vesicle size and endocytic proteins across species, we suggest that this mechanism may represent a broadly conserved principle for sustaining synaptic fidelity (Table 1).

## Methods

### Experimental model and subject details

### Strains

All worm strains were cultured at 22°C under standard conditions on nematode growth medium (NGM) agar plates seeded with *Escherichia coli* OP50 according to standard protocols [32]. Synchronized populations were generated by manual selection of L4-stage larvae and maintained under standard conditions until experimentation. All experiments were conducted on day-1 adult hermaphrodites unless otherwise noted. Mutant and transgenic alleles were backcrossed at least 4 times into the N2 Bristol strains.

### Molecular biology

DNA plasmids were constructed using the Multisite Gateway system (Invitrogen, Waltham, MA, USA) and Gibson assembly protocols [85], with all constructs verified by sequencing. The promoter *snb-1p* (3kb), which drives pan-neuronal expression, was used in rescue experiments. For rescue experiments, the *snb-1* promoter was cloned into modified pCFJ150 vector [86] along with the cDNA encoding wild-type or mutant proteins. Full length UNC-11 (C32E8.10b.1) and HIPR-1 (ZK370.3a.1) cDNA sequences were obtained by PCR amplification from a home-made *C. elegans* cDNA library. Rat Epsin-1 (NP_476477.1) cDNA was amplified from the Addgene 22228 plasmid. Mouse neurofilament-M (NP_032717.2), AP180 (NP_001344699.1), and CALM (NP_666306.2) coding sequences were amplified from a mouse cDNA library. DNA constructs encoding chimeric UNC-11 variants were generated using overlap extension PCR strategy [87] (Table 1).

### Transgenes and germline transformation

Transgenic strains were generated by microinjection (Table 1). Single-copy transgenes were introduced using the Mos1-mediated single-copy insertion (MosSCI) method [86,88]. Briefly, injection mixes containing 60 ng/μl of a plasmid encoding sgRNA targeting mos1 and 40 ng/μl of a repair plasmid carrying the desired gene fragments were injected into the gonads of young adult worms. Transgenic F1 progeny was identified based on fluorescent markers and/or drug resistance, and successful insertions were confirmed by PCR. All transgenic strains were outcrossed at least four times to the N2 background. For genetic crosses, transgenes were introduced into mutant backgrounds using standard mating protocols, and homozygosity was verified by fluorescence and PCR analysis. Extrachromosomal arrays were generated by injecting DNA mixtures containing 10 ng/μl of the target plasmid, 10 ng/μl of a fluorescent co-injection marker, 15 ng/μl of a drug-selection marker, and 65 ng/μl of pBlueScript as carrier DNA. F1 worms exhibiting fluorescence were isolated, and stable lines were established by selecting F2 populations with high transmission rates.

### Worm tracking and locomotion analysis

Worm locomotion was tracked and analyzed as previously described [47,89]. Young adult animals (day 1) were picked to 10 cm NGM agar plates with no bacterial lawn (20 worms per plate) and acclimated for 1 hour at room temperature. Worm

**Table 1.** Key resources table.

| Reagent or resource | Source | Identifier |
|---|---|---|
| **Bacterial** | | |
| *E. coli*: OP50 | CGC | OP50 |
| **Chemicals and Critical Commercial Assays** | | |
| PrimeSTAR GXL DNA Polymerase | TaKaRa | Cat No. R050A |
| PrimeSTAR Max DNA Polymerase | TaKaRa | Cat No. R045B |
| Phusion polymerase | NEB Biolabs | Cat No. M0530S |
| HEPES | ThermoFisher | Cat. No. BP310-500 |
| Agar | Apex | Cat. No. 20-275 |
| 2,3-butanedione monoxime (BDM) | Sigma | Cat. No. B0753 |
| Tetramisole Hydrochloride | Sigma | Cat. No. T1512 |
| QIAquick PCR Purification Kits | QIAGEN | Cat. No. 28104 |
| QIAquick Gel Extraction Kits | QIAGEN | Cat. No. 28704 |
| QIAprep Spin Miniprep Kits | QIAGEN | Cat. No. 27104 |
| 1-palmitoyl-2-oleoyl-sn-glycero-3-phosphatidylcholine (1,2-POPC) | Larodan Inc | Cat. No. 26853-31-6 |
| 1,2-Dioleoyl-sn-glycero-3-phospho-L-serine, sodium salt (POPS-Na) | COATSOME | Cat. No. MS-8181LS |
| L-α-phosphatidylinositol-4,5-bisphosphate (Brain, Porcine) (ammonium salt) (Brain PI(4,5)P2) | Avanti | Cat. No. 840046X-5mg |
| 1,2-dioleoyl-sn-glycero-3-phosphoethanolamine-N-(lissamine rhodamine B sulfonyl) (ammonium salt) (18:1 Liss Rhod PE) | Avanti | Cat. No. 810150-1mg |
| Pacific Blue C5-Maleimide | ThermoFisher | Cat. No. P30506 |
| Alexa Fluor 488 NHS ester | ThermoFisher | Cat. No. A20000 |
| Alexa Fluor 405 NHS ester | ThermoFisher | Cat. No. A30000 |
| Actin-stain 488 Phalloidin | Cytoskeleton | Cat. No. PHDG1-A |
| Muscle Actin >95% pure, Rabbit Skeletal Muscle | Cytoskeleton | Cat. No. AKL95 |
| **Experimental Models: Organisms/Strains** | | |
| Wild type Bristol isolate | CGC | N2 |
| *unc-11(e47) I* | [32,34] | BJH2039 |
| *unc-11(pek217) I* | This study | BJH717 |
| *kyIs673 [sra-6p::eat-4::pHluorin]* | [81] | CX16921 |
| *ttTi5605 II* | This study | BJH2002 |
| *unc-11(pek217) I; pekSi68 [snb-1p::unc-11, neoR(+)] II* | This study | BJH2461 |
| *unc-11(pek217) I; pekSi81[snb-1::unc-11::GFP, neoR(+)] II* | This study | BJH2097 |
| *unc-11(pek217) I; pekSi92 [snb-1p::unc-11ΔAD, neoR(+)] II* | This study | BJH966 |
| *unc-11(pek217) I; pekSi151[snb-1p::unc-11ΔANTH, neoR(+)] II* | This study | BJH2130 |
| *unc-11(pek217) I; kyIs673 [sra-6p::pHluorin]* | This study | BJH2377 |
| *unc-11(pek217) I; pekSi92[snb-1p::unc-11ΔAD, neoR(+)] II; kyIs673 [sra-6p::pHluorin]* | This study | BJH2379 |
| *unc-11(pek217) I; pekSi68[snb-1p::unc-11, neoR(+)] II; kyIs673 [sra-6p::pHluorin]* | This study | BJH2466 |
| *cpx-1(ok1552) unc-11(pek217) I; pekSi92 [snb-1p::unc-11ΔAD, neoR(+)] II* | This study | BJH2150 |
| *cpx-1(ok1552) I* | [82] | KP6176 |
| *unc-11(pek217) I; pekSi449[snb-1p::unc-11ΔAD::mCALM AD, neoR(+)] II* | This study | BJH2533 |
| *unc-11 (pek217); pekSi495[snb-1p::unc-11ΔAD::mAP180 AD, neoR(+)] II* | This study | BJH2609 |

*(Continued)*

**Table 1.** (Continued)

| Reagent or resource | Source | Identifier |
|---|---|---|
| unc-11(pek217) I; pekSi336 [snb-1p::unc-11ΔAD::hipr-1 CD, neoR(+)] II | This study | BJH2301 |
| unc-11 (pek217) I; pekSi344 [snb-1p::unc-11ΔAD::neurofilament-M CD] II | This study | BJH2309 |
| unc-11 (pek217) I; pekSi342 [snb-1p::unc-11ΔAD::epsin-1 CD, neoR(+)] II | This study | BJH2308 |
| unc-11(pek217) I; pekSi450[snb-1p::unc-11ΔAD::hipr-1ΔTHATCH, neoR(+)] II | This study | BJH2518 |
| unc-11(pek217) I; pekSi450[snb-1p::unc-11ΔAD::lifeact, neoR(+)] II | This study | BJH2508 |
| hipr-1(ok1081) III | [82] | RB1102 |
| cpx-1(ok1552) I; tauIs114 [snb-1p::cpx-1::GFP::rab-3] IV | [51] | JSD0374 |
| unc-11 (pek217) I; pekSi92[snb-1p::unc-11ΔAD, neoR(+)] II; tauIs114 [snb-1p::cpx-1::gfp::rab-3] IV | This study | BJH2392 |
| unc-11(pek217) I; pekSi92 [snb-1p::unc-11ΔAD; neoR(+)] II; pekEx203 [rab-3p::cpx-1_ΔCH::rab-3] | This study | BJH1062 |
| pekSi727[snb-1p::hipr-1Δl/LWEQ::unc-11AD, neoR(+)] II; hipr-1(ok1081) III | This study | BJH4061 |
| unc-11(pek217) I, pekSi92[snb-1p::unc-11ΔAD, neoR(+)] II; pekEx195[snb-1p::cpx-1::gfp] | This study | BJH1054 |
| **Plasmid:** | | |
| Plasmid: snb-1p::unc-11ΔAD::unc-54utr | This study | BJP-A234 |
| Plasmid: snb-1p::unc-11::unc-54utr | This study | BJP-A199 |
| Plasmid: snb-1p::unc-11ΔANTH::unc-54utr | This study | BJP-A528 |
| Plasmid: snb-1p::unc-11ΔAD::hipr-1 CD::unc-54utr | This study | BJP-C87 |
| Plasmid: snb-1p::unc-11ΔAD::neurofilament-M CD::unc-54utr | This study | BJP-C88 |
| Plasmid: snb-1p::unc-11ΔAD::epsin-1 CD::unc-54utr | This study | BJP-C89 |
| Plasmid: snb-1p::unc-11ΔAD::hipr-1ΔTHATCH::unc-54utr | This study | BJP-C726 |
| Plasmid: snb-1p::unc-11ΔAD::lifeAct::unc-54utr | This study | BJP-C757 |
| Plasmid: rab-3p::cpx-1_ΔCH::rab-3::unc-54utr | This study | BJP-C705 |
| Plasmid: snb-1p::unc-11ΔAD::mAP180 AD::unc-54utr | This study | BJP-C762 |
| Plasmid: snb-1p::unc-11ΔAD::mCALM AD::unc-54utr | This study | BJP-C761 |
| Plasmid: unc-11AD(305-end)::strepII::his6 | This study | BJP-C551 |
| Plasmid: unc-11FL::strepII::his6 | This study | BJP-C550 |
| Plasmid: unc-11ANTH(1–304)::strepII::his6 | This study | BJP-SY22 |
| **Software and imagining equipment** | | |
| Excel | Microsoft | N/A |
| Adobe Illustrator | Adobe | N/A |
| Prism 9 | GraphPad Prism | N/A |
| RStudio | Posit | N/A |
| WormLab Imaging System | MBF Bioscience, VT, USA | N/A |
| Olympus FV-1000 Confocal Microscope | Olympus Company | N/A |
| SnapGene 5.0 | SnapGene | N/A |
| Igor Pro | Wavemetrics, OR, USA | N/A |
| ColabFold | [83] | N/A |

*(Continued)*

**Table 1.** (Continued)

| Reagent or resource | Source | Identifier |
|---|---|---|
| PyMOL | PyMOL by Schrödinger | N/A |
| ImageJ | [84] | N/A |
| Clampfit | Molecular Devices | N/A |

crawling on the agar surface was recorded for 30 s using the WormLab Imaging System (MBF Bioscience, VT, USA). Animal trajectories were analyzed using a ImageJ/FIJI plugin implementing a particle-tracking algorithm optimized for *C. elegans* [84]. The average speed was determined for each animal.

## VGLUT-pHluorin imaging and analysis

Imaging of VGLUT-pHluorin fluorescence in ASH axons was carried out using *C. elegans* strains expressing the transgene *kyIs673 [sra-6p::eat-4::pHluorin]*, following established protocols [81]. Animals were immobilized with 1 mM tetramisole hydrochloride and placed into custom-fabricated PDMS microfluidic chambers designed to deliver chemical stimuli during fluorescence microscopy [90]. A 500 mM NaCl solution in S. Basal Buffer was freshly prepared each day for stimulation. Imaging was performed using a Leica DMi8 inverted microscope equipped with a Leica PL APO 63×/1.40 NA oil immersion objective and an Andor iXon Life 888 EMCCD camera, controlled via Leica LAS-X software. Worms were allowed to rest in the chamber for 5 min before imaging, and adapted to blue light for 90 s as described previously [81]. Each animal was imaged in no more than three trials, with each session lasting 100 s; a 10-s stimulus was delivered starting at the 30-second mark. Images were acquired at 5 frames per second.

Imaging analysis was performed as previously described [47]. Motion correction was applied to compensate for lateral drift, and datasets with significant z-axis drift were excluded. Axonal regions of interest (ROIs) were manually defined in 3 × 3 pixel blocks, with flanking background ROIs selected for baseline correction. Fluorescence intensities were extracted for both signal and background regions. To correct for photobleaching and background fluctuation, a linear model was fit to the background signal using MATLAB's polyfit function and subtracted from the raw traces. After correcting for bleaching, background-subtracted traces were used to quantify fluorescence changes. Baseline fluorescence was calculated by averaging the first 10 s of each ROI following background correction. Fluorescence decay and $\Delta F/F$ were calculated using standard curve-fitting approaches. All imaging conditions were independently replicated on at least two separate days with fresh buffer and stimulus preparations. Trial numbers per animal and total sample sizes are detailed in the figure legends.

## Electrophysiology

Electrophysiological recordings were performed on Day-1 young adult hermaphrodite worms as described previously [89,91,92]. Animals were immobilized on Sylgard-coated coverslips using tissue adhesive glue (Histoacryl Blue, Braun) and dissected in extracellular solution by making a dorsolateral incision with a sharpened tungsten needle. The gonad and intestines were removed via suction through a glass pipette. To expose the ventral nerve cord and surrounding body wall muscle, the cuticle flap was carefully reflected and secured using glue. Prepared animals were transferred to a fixed-stage upright microscope (BX51WI, Olympus) equipped with a 60× water-immersion objective. The structural integrity of the anterior ventral body wall muscles and ventral nerve cord was confirmed using differential interference contrast microscopy. Whole-cell patch clamp recordings were obtained from ventral body wall muscle cells using fire-polished borosilicate pipettes (2–5 MΩ resistance, World Precision Instruments). Recordings were performed at 20°C, with muscle cells voltage-clamped at −60 mV to monitor postsynaptic currents using an EPC-10 amplifier (HEKA, Germany). The extracellular

recording solution contained (in mM): 150 NaCl, 5 KCl, 1 CaCl$_2$ (or 0.25), 5 MgCl$_2$, 10 glucose, and 10 HEPES, adjusted to pH 7.3 with NaOH and to 330 mOsm with sucrose. The internal pipette solution contained (in mM): 135 Cs methanesulfonate, 5 CsCl, 5 MgCl$_2$, 5 EGTA, 0.25 CaCl$_2$, 10 HEPES, and 5 Na$_2$ATP, titrated to pH 7.2 with CsOH. Evoked excitatory postsynaptic currents (EPSCs) were elicited by a 0.4 ms, 30 μA electrical pulse delivered through a ~2 MΩ borosilicate pipette positioned near the ventral nerve cord, using a stimulus isolator (A365, WPI). Series resistance was compensated by 70% during evoked EPSC recordings. Data were acquired at a sampling rate of 10 kHz with Patchmaster software (HEKA) and low-pass filtered at 2 kHz. All reagents were obtained from Sigma. Sample sizes for each experimental condition are provided in the figure legends.

## Transmission electron microscope for visualizing synaptic vesicle

High-pressure freezing and transmission electron microscopy were used to examine synaptic ultrastructure. Approximately 10 adult hermaphrodites were rapidly loaded into a 100 μm high-pressure freezing chamber filled with a bacterial suspension. Worms were flash-frozen at approximately −180°C using a Leica EMPact 2 system (Leica Microsystems, Vienna). Frozen samples underwent freeze substitution in a Leica EM AFS2 unit with a fixative solution containing 1% osmium tetroxide and 0.1% uranyl acetate. After freeze substitution, specimens were rinsed three times with pure acetone, then gradually infiltrated and embedded in Eponate12 resin (Ted Pella, , Redding, CA). Serial ultrathin sections (35–40 nm) were prepared using a Leica EM UC7 ultramicrotome, stained with uranyl acetate and lead citrate, and imaged at 120 kV using a Talos L120C transmission electron microscope (Thermo Fisher Scientific, Waltham, MA). Digital images were captured with a Ceta 16M CMOS 4k × 4k camera (Thermo Fisher Scientific). Synaptic vesicle (SV) counts were obtained from individual synapse profiles. Each profile corresponded to a single section passing through the dense projection of the synapse.

## Recombinant protein production, fluorescence labeling, and droplet formation

Recombinant proteins were expressed as C-terminal StrepTagII- and His6- tagged fusion proteins in the BL21(DE3) *E. coli* strain. Protein purification was carried out using published protocols [47,93]. Briefly, bacterial cultures were grown in Luria Broth at 37°C and induced with 0.2 mM isopropyl β-D-1-thiogalactopyranoside (IPTG) when the optical density at 600 nm (OD$_{600}$) reached 1.0. Cells were harvested by centrifugation and lysed with a microfluidizer in lysis buffer containing 20 mM HEPES (pH 8.0), 300 mM NaCl, and 15 mM imidazole. Proteins were purified using Ni–NTA agarose (Qiagen, Valencia, CA) and eluted with lysis buffer supplemented with 250 mM imidazole. Purified proteins were dialyzed against 20 mM HEPES (pH 7.7) and 150 mM NaCl, and stored at 4°C in the presence of 1 mM dithiothreitol (DTT).

Pacific Blue C5-Maleimide, Alexa Fluor 488 NHS ester, and Alexa Fluor 405 NHS ester (Thermo Fisher Scientific) were used for fluorescence labeling. For Pacific Blue labeling, UNC-11 was incubated overnight at 4°C with a 10-fold molar excess of dye in 20 mM HEPES (pH 7.7), 150 mM NaCl, and 5 mM Tris(2-carboxyethyl)phosphine (TCEP). For Alexa Fluor 488 and 405 labeling, proteins were incubated with a 3-fold molar excess of Alexa Fluor 488 NHS ester or Alexa Fluor 405 NHS ester in the same buffer at 4°C. Labeled proteins were dialyzed for 4 hours against a 1,000-fold volume of HEPES buffer (20 mM HEPES, pH 7.7, 150 mM NaCl) to remove unbound dye.

For droplet formation assays, unlabeled and Alexa Fluor 488–labeled UNC-11 variants were mixed at a 9:1 molar ratio. Proteins at the indicated concentrations were incubated for 30 min at room temperature in 20 mM HEPES (pH 7.7) containing 10% w/v polyethylene glycol (PEG 3350) and NaCl at concentrations ranging from 50 to 200 mM.

## Liposome preparation

Lipids were stored in chloroform at −20°C. POPC (1-palmitoyl-2-oleoyl-sn-glycero-3-phosphocholine; designated PC) was obtained from Larodan DOPS (1,2-dioleoyl-sn-glycero-3-phospho-L-serine; designated PS) was purchased from NOF

America Corporation. PI(4,5)P2 (brain L-α-phosphatidylinositol-4,5-bisphosphate, $PIP_2$) and 18:1 Liss Rhodamine PE (Rhod-PE) were obtained from Avanti Polar Lipids. Lipids were mixed in glass tubes at the desired ratios. PC liposomes were composed of 98% PC and 2% Rhod-PE. PIP2 liposomes contained 71% PC, 25% PS, 2% PI(4,5)P2, and 2% Rhod-PE. The lipid mixtures were dried under a gentle stream of nitrogen gas for 30 min, and any remaining solvent was removed under vacuum using a Labconco FreeZone 2.5 lyophilizer for 2 h. The dried lipid films were rehydrated in HEPES buffer (20 mM HEPES, 150 mM NaCl, pH 7.7) and briefly vortexed to homogenize the suspension. The resulting lipid dispersions were extruded 11 times through 50 nm pore-size polycarbonate membranes using a Mini Extruder (Avanti Polar Lipids) to generate unilamellar vesicles.

## Actin preparation and labeling

Rabbit skeletal muscle actin and pyrene-labeled actin were obtained from Cytoskeleton Monomeric G-actin was prepared by resuspending 1 mg of actin in 1 ml General Actin Buffer (5 mM Tris-HCl, pH 8.0; 0.2 mM $CaCl_2$) supplemented with 0.2 mM ATP. The solution was incubated on ice for 1 h to depolymerize actin oligomers formed during storage and centrifuged at 14,000 rpm for 30 min at 4°C. The supernatant was transferred to a new microfuge tube, yielding G-actin at approximately 20 μM.

Pyrene G-actin was prepared from the Actin Polymerization Biochem Kit (Cytoskeleton). A 5 μl frozen aliquot of pyrene actin was diluted with 225 μl General Actin Buffer, mixed gently, and incubated on ice for 1 h to depolymerize actin oligomers. The solution was centrifuged at 14,000 rpm for 30 min at 4°C, and the resulting supernatant contained pyrene G-actin at a final concentration of approximately 10 μM.

F-actin was prepared from 250 μg skeletal muscle actin (Cytoskeleton). Actin was resuspended to 1 μg/μl in 250 μl General Actin Buffer (5 mM Tris-HCl, pH 8.0; 0.2 mM $CaCl_2$) and incubated on ice for 30 min. Polymerization was initiated by adding 25 μl of 10× Actin Polymerization Buffer (500 mM KCl, 20 mM $MgCl_2$, and 10 mM ATP) and incubating at room temperature (22°C) for 1 hour. F-actin was stained with Actin-stain 488 Phalloidin (Cytoskeleton, ) by incubating samples containing 3.5 μM fluorescent Phalloidin and 10 μM F-actin overnight at 4°C.

## Fluorescence microscopy and imaging analyses

Imaging chambers were assembled using a 35×50 mm glass slide and a 25×25 mm piece of parafilm. Glass slides were soaked in HEPES buffer (20 mM HEPES, pH 7.7, 150 mM NaCl) for 30 min and dried with light-duty tissue wipers (VWR International). A 15×15 mm square was cut into the parafilm, which was pressed onto the glass slide to form a well. Ten microliters of protein solution were placed in the chamber and sealed with a 22×22 mm coverslip to prevent evaporation during imaging.

Fluorescence recovery after photobleaching (FRAP) and droplet dynamics was performed using an Olympus FV1000 laser-scanning confocal microscope equipped with an Olympus UPLSAPO 60×/1.4 NA oil-immersion objective at 5× zoom. Pacific Blue and Alexa Fluor 405 were excited using a 405 nm argon laser, Alexa Fluor 488 using a 488 nm argon laser, and Rhodamine-PE using a 559 nm diode-pumped solid-state laser. Protein droplets were formed by incubating 15 μM UNC-11 AD (20% labeled with Alexa Fluor 405) or 10 μM full-length UNC-11 (20% labeled with Pacific Blue) in HEPES buffer (20 mM HEPES, pH 7.7, 150 mM NaCl) containing 10% w/v PEG 3350 for 30 min. Prior to photobleaching, an image was acquired to record Alexa Fluor 488 fluorescence using 0.5% power from the 488 nm laser. A circular ROI encompassing a single protein droplet was bleached using full laser power, and fluorescence recovery was recorded at 5 s intervals for 250 s. Images were analyzed in ImageJ [74]. Fluorescence intensities were normalized to pre-bleach levels and are reported as mean±SEM over time.

For visualizing actin incorporation, 2 μM Alexa Fluor 488–labeled G-actin (50% labeled) was added, and fluorescence images were collected every 5 min for 60 min (UNC-11 AD droplets) or for 20 min (full-length UNC-11 droplets). Fluorescence intensity within droplets was quantified using ImageJ and normalized to the final time point. Data from four independent experiments are reported as mean±SEM. To monitor actin polymerization initiated from UNC-11 droplets, unlabeled

G-actin (2 µM) was used together with 0.2 µM Alexa Fluor 488–phalloidin. Imaging began 3 min after addition of actin and phalloidin, and images were taken every 3 min for 48 min.

To monitor actin polymerization, pyrene-labeled G-actin (2 µM) was incubated in HEPES buffer containing 10% w/v PEG 3350 and 0.2 mM ATP, with or without UNC-11 droplets. Pyrene fluorescence was excited at 350 nm, and emission spectra were collected every minute using a Chirascan CCD Emission Fluorometer (Applied Photophysics, UK). Fluorescence intensity at 405 nm was normalized to the final value of the control sample without droplets.

For protein droplet-actin association assays, UNC-11 full-length droplets (20% labeled with Pacific Blue) and F-actin (labeled with Alexa Fluor 488-phalloidin) were prepared as described above and mixed to final concentrations of 10 µM UNC-11 and 2 µM actin. Samples were incubated for 30 min at room temperature. Fluorescence z-stacks were collected at 0.5 µm intervals using the Olympus FV1000 microscope with a 63×/1.4 NA objective. For assays combining protein droplets, F-actin, and liposomes, 0.5 mM total lipids were added to droplet-actin mixtures and incubated for 30 min at room temperature.

## Statistics

Student *t* test was used for single pairwise comparisons. For comparisons involving more than two groups, one-way ANOVA was applied. When significant differences were detected, Tukey's Honestly Significant Difference (HSD) test was used for post hoc pairwise comparisons. Categorical data were analyzed using Fisher's exact test. Statistical significance was defined as $p < 0.05$ (*$p < 0.05$, **$p < 0.01$, ***$p < 0.001$). Statistical analyses and data visualization were performed using GraphPad Prism 9, RStudio, Clampfit (Molecular Devices), ImageJ [84], FluoView (Olympus), and Microsoft Excel.

## Supporting information

**S1 Fig. Two *unc-11* mutant alleles exhibit identical synaptic transmission defects. (A)** Schematic of the *Caenorhabditis elegans ap180 unc-11* gene showing the exons of its isoform b transcript (C32E8.10b.1) and the positions of two mutant alleles. The *e47* allele deletes 210 bp across exons 1 and 2 (shaded box). The *pek217* allele, generated by CRISPR-Cas9, removes most of *unc-11*, including part of exon 3, all of exons 4–7, and connecting introns. **(B)** Representative evoked EPSC traces (*left*) and summary data for the amplitude of evoked EPSCs (*right*) are shown. **(C)** Representative traces (*left*) and summary data of endogenous EPSC frequency (*middle*) and amplitude (*right*) for indicated genotypes. Data are presented as mean ± SEM; the number of worms is indicated in the bar graphs. Error bars represent SEM. Statistical analysis: one-way ANOVA with Tukey's HSD post hoc test. Significance levels are denoted *** $p < 0.001$. The data underlying this figure are provided in S8 Data.
(TIF)

**S2 Fig. UNC-11 AD alone fails to rescue locomotion and synaptic transmission in *unc-11* mutant worms. (A)** Schematic of UNC-11 AD (residues 305–546) lacking the ANTH domain (residues 1–304). UNC-11 AD is expressed in *unc-11* mutant worms via a single-copy transgene under the pan-neuronal *snb-1p* promoter. "AD" indicates *unc-11* mutants expressing UNC-11 AD without ANTH. Summary data for locomotion rates **(B)**, evoked EPSC amplitude **(C)**, and endogenous EPSC frequency and amplitude **(D)** are shown. The number of worms per genotype is indicated. Data are shown as mean ± SEM. Statistical analysis: one-way ANOVA with Tukey's HSD post hoc test. "n.s." indicates no significance. The data underlying this figure are provided in S9 Data.
(TIF)

**S3 Fig. Conserved role of ADs from mouse AP180, CALM, and *Caenorhabditis elegans* UNC-11 in regulating synaptic vesicle size. (A)** Schematics of chimeric proteins where UNC-11ΔAD is fused to the assembly domain of mouse AP180 (mAP180 AD, residues 287–902) or mouse CALM (mCALM AD, residues 287–661). These chimeras are expressed in *unc-11* mutant worms via single-copy transgenes under the pan-neuronal *snb-1p* promoter. "mAP180 AD"

and "mCALM AD" refer to *unc-11* mutants expressing UNC-11 chimeras containing mAP180 AD and mCALM AD, respectively. **(B)** Scatter dot plots show synaptic vesicle diameter summary data. Mean values are indicated by the horizontal lines on the graph in the graph. Each data point represents one synaptic profile. "n.s." indicates no significance. *** $p < 0.001$ (one-way ANOVA, Tukey's HSD post hoc test). The data underlying this figure are provided in S10 Data.
(TIF)

**S4 Fig. Synaptic transmission at UNC-11ΔAD synapses is insensitive to *cpx-1* mutation at 0.25 mM external calcium. (A)** Representative traces and **(B)** summary data of endogenous EPSC frequency (*left*) and amplitude (*right*) for the indicated genotypes. "ΔAD": *unc-11* mutant worms expressing UNC-11ΔAD in neurons, and "ΔAD, *cpx-1* mut.": ΔAD worms also carrying the *cpx-1(ok1552)* deletion allele. Data are presented as mean ± SEM. Unpaired Student *t* test with Welch's test (two-tailed); n.s., not significant. The data underlying this Figure are provided in S11 Data.
(TIF)

**S5 Fig. HIPR-1 actin-binding domain controls vesicle size and can be replaced by the UNC-11 AD. (A)** Schematic of the HIPR-1 protein showing the N-terminal ANTH domain and C-terminal actin-binding THATCH domain (yellow). The *hipr-1(ok10181)* deletion mutation truncates the protein after partially removing the ANTH domain. The predicted AlphaFold structure of HIPR-1 is shown below, with the THATCH motif highlighted in yellow. "N" and "C" indicate termini. **(B)** Representative traces (*top*) and summary data (*bottom*) of endogenous EPSC frequency and amplitude for the indicated genotypes. Data are presented as mean ± SEM and were analyzed by unpaired Student *t* test; *** $p < 0.001$; * $p < 0.05$;. **(C)** Transmission electron micrograph of a *hipr-1* mutant synapse showing dense projections (orange) and an endosome-like structure (yellow arrow). **(D)** (*Left*) Summary data of average vesicle diameter per synaptic profile. *** $p < 0.001$ (unpaired Student t test). Each data point represents one synaptic profile, and mean values are indicated by horizontal lines on the graph. (*Right*) Cumulative distribution of vesicle diameters for the indicated genotypes. **(E)** The fraction of synapses containing endosome-like structures (ELS) for the indicated genotypes. Fisher's exact test was used to analyze categorical variables, and exact p-values are denoted. **(F)** Schematics of HIPR-1 chimeras. The HIPR-1ΔTHATCH::AD variant combines HIPR-1 lacking the THATCH domain (deletion of 207 amino acids) with the UNC-11 assembly domain. ΔTHATCH::AD was expressed in *hipr-1* mutant worms as a single-copy transgene driven by the pan-neuronal *snb-1p* promoter. **(G)** Excitatory postsynaptic currents (EPSCs) recorded at the neuromuscular junction. Summary data show endogenous EPSC frequency (*left*) and amplitude (*right*). Data are presented as mean ± SEM, with the number of worms analyzed indicated in the bar graphs. Statistical analysis was performed using one-way ANOVA followed by Tukey's HSD post hoc test. * $p < 0.05$; *** $p < 0.001$; n.s., not significant. The data underlying this figure are provided in S12 Data.
(TIF)

**S6 Fig. UNC-11 and its disordered assembly domain (AD) form protein condensates with fluid properties. (A)** Representative fluorescence images of protein droplets formed by recombinant UNC-11 AD at varying protein concentrations (5 and 10 μM) and ionic strengths (100–200 mM NaCl). The buffer contained 20 mM HEPES (pH 7.7) and 10% w/v PEG. Ten percent of UNC-11 AD was labeled with Alexa Fluor 488 for fluorescence imaging. Scale bar, 10 μm. **(B)** Fluorescence recovery after photobleaching (FRAP) of condensates formed by full-length UNC-11 (UNC-11 FL) and UNC-11 AD. Proteins (10 μM) were incubated in buffer containing 150 mM NaCl, 20 mM HEPES (pH 7.7), and 10% w/v PEG. FRAP was performed using an Olympus FV1000 confocal microscope with a 60×, 1.4 NA oil-immersion objective (5× zoom). A 488 nm argon laser was used for photobleaching. Scale bar, 1 μm. **(C)** Fusion and re-rounding of UNC-11 AD condensates (20 μM) in HEPES buffer (200 mM NaCl, 10% w/v PEG, pH 7.7). Scale bar, 2 μm. **(D)** Fluorescence images showing aggregation of recombinant UNC-11 ANTH (ΔAD) at 10 μM in buffer containing 150 mM NaCl, 20 mM HEPES (pH 7.7), and 10% w/v PEG. Ten percent of UNC-11 ANTH was labeled with Alexa Fluor 488. Scale bar, 10 μm.
(TIF)

**S7 Fig. UNC-11 condensates enrich monomeric actin and interact with actin filaments. (A)** UNC-11 AD condensates enrich monomeric G-actin. (*Upper*) A schematic cartoon depicts the model in which monomeric G-actin (magenta) is recruited and enriched into protein droplets (yellow). (*Lower*) Representative fluorescence images of G-actin (50% labeled with Alexa Fluor 488) and UNC-11 AD (20% labeled with Alexa Fluor 405) taken at 5-min intervals for 60 min after adding G-actin (2 µM) to UNC-11 AD (10 µM). Normalized G-actin fluorescence intensity within droplets was quantified over time and plotted (*right*). Data were collected from four independent experiments and are presented as mean ± SEM. **(B)** Full-length (FL) UNC-11 condensates also enrich monomeric G-actin. *Left:* Representative fluorescence images of G-actin (50% labeled with Alexa Fluor 488) and UNC-11 FL (20% labeled with Alexa Fluor 405) taken at 5-minute intervals for 20 min after adding G-actin (2 µM) to UNC-11 AD (10 µM). *Right:* Normalized G-actin fluorescence intensity within droplets was quantified over time and plotted. Data were collected from four independent experiments and are presented as mean ± SEM. **(C)** UNC-11 condensates associate with F-actin. Representative fluorescence images show F-actin (Alexa Fluor 488–phalloidin, yellow) and protein condensates of UNC-11 AD (*top*) or UNC-11 FL (*bottom*, magenta). F-actin was assembled from 2 µM G-actin followed by addition of UNC-11 variants (10 µM). Scale bar, 5 µm. The data underlying this figure are provided in S13 Data.
(TIF)

**S8 Fig. UNC-11 requires membrane-binding interactions to connect condensates to liposomes. (A)** Representative fluorescence images of samples containing F-actin, PIP2(2%) liposomes, and UNC-11 AD that lacks the membrane-binding ANTH domain. UNC-11 AD is labeled with Alexa Fluor 405 (magenta), F-actin with Alexa Fluor 488-phalloidin (yellow), and liposomes with Rhodamine PE (2%, cyan). Liposomes contained 500 µM total lipids (2% PIP2, 25% PS, 71% PC, and 2% Rhodamine-PE). Scale bar, 5 µm. **(B)** Representative fluorescence images of samples with F-actin, PC liposomes (98%PC, 2%Rhodamine-PE), and UNC-11 FL. UNC-11 binds liposomes carrying anionic phospholipid but not PC-only liposomes. UNC-11 FL was labeled with Pacific Blue (magenta), F-actin with Alexa Fluor 488-phalloidin (yellow), and liposomes with Rhodamine PE (2%, cyan). Liposomes contained 500 µM total lipids (98% PC and 2% Rhodamine-PE). Scale bar, 5 µm. The data quantified from the images in this figure are presented in Fig 7, and the source data are provided in S7 Data.
(TIF)

**S1 Data. Source data for Fig 1C, 1D, 1F, and 1G.**
(XLSX)

**S2 Data. Source data for Fig 2B–2E.**
(XLSX)

**S3 Data. Source data for Fig 3B–3D.**
(XLSX)

**S4 Data. Source data for Fig 4A, 4B, and 4E.**
(XLSX)

**S5 Data. Source data for Fig 5C and 5D.**
(XLSX)

**S6 Data. Source data for Fig 6A–6C.**
(XLSX)

**S7 Data. Source data for Fig 7B and 7C.**
(XLSX)

**S8 Data. Source data for S1B and S1C Fig.**
(XLSX)

**S9 Data. Source data for S2B–S2D Fig.**
(XLSX)

**S10 Data. Source data for S3B Fig.**
(XLSX)

**S11 Data. Source data for S4B Fig.**
(XLSX)

**S12 Data. Source data for S5B–S5E, and S5G Fig.**
(XLSX)

**S13 Data. Source data for S7A Fig.**
(XLSX)

## Acknowledgments

We thank Dr. Cori Bargmann for worm strains. Electron microscopy data were generated using the Fred Hutchinson Cancer Center Electron Microscopy Shared Resource (EMSR).

## Author contributions

**Conceptualization:** Yu Wang, Lanxi Wu, Jihong Bai.

**Data curation:** Yu Wang, Lanxi Wu, Lin Zhang, Yongming Dong, Aaradhya Pant, Yan Liu.

**Formal analysis:** Yu Wang, Lanxi Wu, Lin Zhang, Yongming Dong, Aaradhya Pant, Yan Liu, Jihong Bai.

**Funding acquisition:** Jihong Bai.

**Investigation:** Yu Wang, Lanxi Wu, Lin Zhang, Yongming Dong, Aaradhya Pant, Yan Liu, Jihong Bai.

**Methodology:** Yu Wang, Lanxi Wu, Lin Zhang, Yongming Dong, Aaradhya Pant, Jihong Bai.

**Project administration:** Jihong Bai.

**Resources:** Jihong Bai.

**Software:** Yongming Dong.

**Supervision:** Jihong Bai.

**Validation:** Yu Wang, Lanxi Wu, Lin Zhang, Yongming Dong, Aaradhya Pant, Yan Liu.

**Visualization:** Yu Wang, Lanxi Wu, Aaradhya Pant.

**Writing – original draft:** Yu Wang, Jihong Bai.

**Writing – review & editing:** Yu Wang, Lanxi Wu, Lin Zhang, Yongming Dong, Aaradhya Pant, Yan Liu, Jihong Bai.

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
