## [Editor Report · Decision Letter 0]

9 Jun 2025

Dear Dr Bai,

Thank you for submitting your manuscript entitled "Synaptic Vesicle Size and Release Fidelity Controlled by AP180 Assembly Domain" for consideration as a Research Article by PLOS Biology.

Your manuscript has now been evaluated by the PLOS Biology editorial staff, as well as by an academic editor with relevant expertise, and I am writing to let you know that we would like to send your submission out for external peer review.

We think that your article would best fit our Short Reports format, given its current scope. This means that, prior to publication, you will need to reduce the number of figures to a maximum of four, for example by combining figures or by moving some elements into the supplement. But it's not necessary for you to make these formatting changes at this stage (and also note that if your paper is substantially expanded during a revision, we could change the format back to a Research Article).

Once your full submission is complete, your paper will undergo a series of checks in preparation for peer review. After your manuscript has passed the checks it will be sent out for review. To provide the metadata for your submission, please Login to Editorial Manager (https://www.editorialmanager.com/pbiology) within two working days, i.e. by Jun 11 2025 11:59PM.

Kind regards,

Taylor

Taylor Hart, PhD,

Associate Editor

PLOS Biology

thart@plos.org

---

## [Decision Letter · Decision Letter 1]

1 Aug 2025

Dear Dr Bai,

Thank you for your patience while your manuscript "Synaptic Vesicle Size and Release Fidelity Controlled by AP180 Assembly Domain" was peer-reviewed at PLOS Biology. It has now been evaluated by the PLOS Biology editors, an Academic Editor with relevant expertise, and by several independent reviewers.

In light of the reviews, which you will find at the end of this email, we would like to invite you to revise the work to thoroughly address the reviewers' reports.

As you will see, the reviewers found the topic interesting and the data compelling. However, they also noted areas where the proposed mechanism is not fully supported by the reported data and suggested additional experiments that could address these points. In addition, they noted areas requiring clarifications or better definition of the limitations of the study. You should thoroughly revise your manuscript to address in full all the reviewers' points.

In addition, as mentioned previously, we think that your article might be best suited for our Short Report format. In preparing your revision, please ensure that there are four or fewer main figures, and select this format when you re-submit. However, if these revisions substantially increase the scope of your paper, we could consider the revised version as a full Research Article.

Given the extent of revision needed, we cannot make a decision about publication until we have seen the revised manuscript and your response to the reviewers' comments. Your revised manuscript is likely to be sent for further evaluation by all or a subset of the reviewers.

**IMPORTANT - SUBMITTING YOUR REVISION**

*Re-submission Checklist*

*Published Peer Review*

*PLOS Data Policy*

*Blot and Gel Data Policy*

Sincerely,

Taylor

Taylor Hart, PhD,

Associate Editor

PLOS Biology

thart@plos.org

REVIEWS:

Reviewer's Responses to Questions

Reviewer #1: SUMMARY

To maintain neuronal function, synaptic vesicles are locally recycled at synapses. AP180, known as UNC-11 in C. elegans, binds synaptobrevin, clathrin, and membranes to ensure the fidelity and efficiency of synaptic vesicle generation. Membrane binding is mediated by binding of the ANTH domain to PIP2. Binding to AP2 and clathrin is mediated by the less well-understood assembly domain (AD), described in this manuscript. Here, the authors focus on the roles of the unstructured nature and vesicle association of the assembly domain in driving membrane curvature and make the following claims:

* The AD inhibits neurotransmission by decreasing synaptic vesicle diameter.

* AP180 interacts with actin to regulate vesicle size.

* Large vesicles, resulting from deletion of the AD, evade regulation by complexin which causes an increase in vesicle fusion.

These results describe a novel mechanism for the role of AP180 in vesicle biogenesis. The first claim is well supported by the data. The following two claims need to be presented with appropriate caveats, as spelled out below. Alternatively, the authors could provide a couple of additional experiments.

Major critique

1) The authors present a simple linear model for the inhibitory function of the AD domain: the AD decreases vesicle diameter and large vesicles evade inhibition of fusion by complexin. In support of this model, a double mutant between AD deletion and complexin has the same elevated mini frequency as either single mutant. However, beyond this observation the single mutants do not phenocopy each other. Specifically, complexin mutants have decreased synaptic vesicle number and dramatically decreased evoked responses. However, deletion of AD results in no significant decrease in synaptic vesicle number and an increase in evoked response. Thus, the increase in synaptic transmission seen in AD deletion is unlikely to be explained simply as large vesicles evading regulation by complexin, otherwise the mutants should look more like complexin mutants. These caveats should be explicitly spelled out in the text.

2) Elevations in mini frequency at calcium levels over 1mM can be challenging to measure. It is possible that the lack of an increase in the cpx-1 unc-11(ΔAD) double mutant is the result of a ceiling effect; in other words, at 1mM calcium complexin mutants are already fusing at their maximum rate and can't be further increased. Adding an additional data point at 0.25mM or 0.5mM external calcium would help support the model presented.

3) The experiments supporting the interaction with actin as key to AP180 function are:

First, deleting the actin-binding THATCH domain from the HIPR-1 portion of the UNC-11::HIPR-1 chimera abrogates rescue.

Second, replacing the AD of UNC-11 with an actin-binding domain results in a fully functional UNC-11 protein.

There are two possible explanations: either UNC-11 binds actin by binding HIPR-1, or the AD is itself an actin binding module.

Two experiments could further tease which model is correct. 1) The UNC‑11(ΔAD)::Lifeact construct could be put into a hipr-1 mutant background. 2) a hipr-1 mutant with hipr-1(ΔTHATCH)::UNC-11-AD could be tested for rescue. The first experiment would test if UNC-11s role in actin binding is simply via HIPR-1. If this rescues, the AD domain most likely functions to recruit HIPR-1 which in turn binds actin. The second experiment would test the novel idea that UNC-11 itself can bind actin. Of course, there are numerous reasons that these experiments might not work, but success in either of them would go a long way towards establishing the details of link between UNC-11 and actin. Adding them would strengthen the manuscript. Absent new experiments, we suggest a thoughtful explanation of the caveats of the current experiments. As currently written it's easy for the reader to be misled into thinking that UNC-11 interacts directly with actin - a point not supported by the experiments.

Minor critique and typographical errrors

1) The isoform of unc-11 being used is never mentioned (even in supplemental Figure 1). The gene structure looks like isoform b. Please indicate which isoform is shown in Fig S1.

2) There is a general lack of consistency with mentioning statistical significance in the text (e.g lines 132 and 163 could say "NS"). In keeping with this, often a comparison is made in the text but no accompanying significance bar is present in the referenced graph (e.g. line 234 compares UNC-11∆AD::NfM-CD to unc-11 but this is not highlighted on graph). Significance indicators can make a graph busy, so I understand why the author omits some comparisons, but don't quite understand why others are included but not referenced. I would rather the author add omitted but referenced significance bars, rather than remove included but unreferenced bars, because in the future, someone may appreciate the comparison, even if it was not needed to support the story presented in this text.

3) FIGURE 3C and FIGURE S5D. The distribution of SV diameters could be more clearly presented. Currently the Y-axis is the total number of vesicles, which simply reflects the number of N2, unc‑11 and delta-D profiles counted. The data should be normalized so the y-axis is 'fraction of total vesicles" in each bin. Moreover, the histograms are overlapping and occlude the other distributions. Making them transparent or "skylines" would facilitate comparisons. A better alternative is to show the size distribution as a cumulative plot rather than binning the data, so that every data point is shown.

4) Mini amplitudes. One might also consider a cumulative plot for mini amplitudes, which is a parallel readout for the synaptic vesicle diameter - the shape of the cumulative plots would be similar.

5) Be consistent between "supplemental" and "supplementary" when referring to supplemental figures.

6) Line 90: "RESULT" should read "RESULTS"

5) Line 103: "Supplemental Figure 1B-C" might just refer to "1C".

6) Lines 130-131: "UNC-11∆AD: 37 ± 2 pA; unc-11 mutant: 31 ± 2 pA; not significant; Figure 1E". This specific comparison is not highlighted with a significance value on graph, whereas others are indicated but not mentioned in the text.

7) Lines 174-175: full-length UNC-11 data should be in Figure 3, not supplemental. It would only add one group to the scatter plot and only one micrograph picture.

8) Line 212: "…C-terminal helix (ΔCH; Figure 4C)". Did you mean "Central Helix"? This is often referred to as the CH, and is responsible for SNARE binding. There is a C-terminal amphipathic helix, but this is thought to interact with membrane.

9) Line 271: "Figure 6 lower" should read "Figure 6a lower"

10) Line 316: "The AD is intrinsic disorder" should read "The AD is intrinsically disordered"

11) Lines 375-376: "Mean values are indicated in the graph" but don't seem to be.

12) Figure 3b: are the numbers indicated in bar chart the number of profiles counted?

13) Figure 5d: half of chart is redundant with 3d.

14) Figure 6a: include label for grey portion of unc11deltaAD.

15) Figure 6c: consider showing statistics for lifeact to N2 comparison.

16) Supplemental Figure 4a: add label to unc-11 ANTH.

17) Supplemental Figure 5a: Here color is used to highlight the THATCH domain. In Figure 1a the default color indicates confidence probably to highlight that is it disordered, but are not indicated in Figure 1a. I don't mind, but consider being consistent.

18) Supplemental Figure 5d: like Figure 3c, make y-axis "fraction of vesicles" to normalize the data.

19) Supplemental Figure 5e: instead of "EL per synapse" like 3d and 5d, make this "fraction of synapses with ELS". This is what the figure legend suggests you are presenting.

Reviewer #2: This is a thoughtful, careful study that addresses an interesting question, the relationship between synaptic vesicle size (due to alterations in endocytic recycling) and the properties of release. All of these molecules have been studied before but not with regard to this important question. The information about the role of the UNC11 assembly domain is also entirely new. The results are novel, suggesting that although other domains of UNC11/AP180 are required for release, the assembly domain both inhibits release and restrains vesicle size.

The data are also compelling. The phenotypes (in the AD mutant) of increased release (evoked and spontaneous) as well as increased vesicle size are clear and convincing. The inability to rescue by replacement with the intrinsically disordered region of an unrelated protein also provides a good control for the replacements by HIPR and epsin that do rescue. In addition to the rescued size, the increased regularity of SV size (Fig. 5D) is particularly compelling. The rescue by fusion to Lifeact also provides strong evidence of a requirement for interaction with actin. It is not surprising that this fusion lowers release below WT since the interaction would lack all the regulation of the endogenous proteins. The rescue of amplitude brings the mutant back to WT, which is reassuring but may also be fortuitous. Regardless, the role of actin seems clear, and the rescue of both frequency and size is reassuring.

There are a few minor points that would benefit from clarification. In the complexin experiment (Fig. 4D), it would help to know that rescue with WT does not work. Clearly, the endogenous complexin does not rescue but it would help to show that the introduced complexin also does not work in the absence of the UNC11 AD. The text describing Fig. S4 should also be revised: UNC11�AD was not replaced; rather, the AD from AP180 was added.

However, the central question is still whether the effect on vesicle size simply correlates with the increased release or causes it. The various rescuing constructs all fix both size and release but maybe these are independent effects. Alternatively, altered vesicle composition might account for the changes in release. Loss of AP180 impairs VAMP2 trafficking but what does the assembly mutant do to VAMP2 or other vesicle proteins? It would also help to show that other mutations affecting SV size (affecting other aspects of the endocytic machinery) all have the same effect. Additional evidence to support a causal role for the effect of vesicle size on release would strengthen this excellent study. I support publication in PLOS Biology but with additional data (or consideration if evidence already exists) that addresses this question.

---

## [Decision Letter · Decision Letter 2]

8 Jan 2026

Dear Dr Bai,

Thank you for your patience while we considered your revised manuscript "Synaptic Vesicle Size and Release Fidelity Controlled by AP180 Assembly Domain" for publication as a Research Article at PLOS Biology. This revised version of your manuscript has been evaluated by the PLOS Biology editors, the Academic Editor, and the original reviewers.

Based on the reviews, we are likely to accept this manuscript for publication, provided you satisfactorily address the remaining point raised by Reviewer 2. Please also make sure to address the following data and other policy-related requests. In addition, we have discussed the format of your paper and agree to publish it as a Research Article, rather than as a Short Report.

-----------

IMPORTANT: Please ensure that you address the following editorial points in your next revision:

**Title:

-- We suggest to tweak your title to provide an indication of AP180's function and specify the study species. Is this alternative acceptable to you?

“Endocytic protein AP180 Assembly Domain regulates synaptic vesicle size and release fidelity in C. elegans”

**Financial disclosure statement:

-- Please add links to the funding agencies in the Financial Disclosure statement in the manuscript details.

Data availability:

-- Thank you for providing the underlying data in the supplement and in your Google Drive repository. Assuming that the files in the supplement and the Drive files are the same, please modify your Data Location statement to reflect that the files are available in both places.

-- We noticed that the caption for S4 Data says that it includes data from Fig. 4A and C, but looking inside the file it seems to contain data from Fig. 4A,B, and E. Please address the inconsistency.

**Abstract

-- Please include the study species in your abstract.

**Code Policy

-- Per journal policy, if you have generated any custom code during the course of this investigation, please make it available without restrictions. Please ensure that the code is sufficiently well documented and reusable, and that your Data Statement in the Editorial Manager submission system accurately describes where your code can be found. More information on our Code Policy, what and how to share can be found here: https://journals.plos.org/plosbiology/s/code-availability

-----------

We expect to receive your revised manuscript within two weeks.

*Published Peer Review History*

*Press*

Sincerely,

Taylor

Taylor Hart, PhD,

Associate Editor

thart@plos.org

PLOS Biology

Reviewer remarks:

Reviewer #1: I noted three critiques in the previous submission:

(1) Complexin. The authors demonstrated that CALM mutants lacking the Accessory Domain (AD) led to enlarged vesicles that exhibited high rates of spontaneous fusion. They noted that the complexin mutants lacking a membrane interaction domain also exhibited high rates of spontaneous fusion and concluded that the large vesicles bypass the complexin fusion clamp. However, the UNC-11-delta-AD mutants did not fully phenocopy complexin mutants which exhibit decreases in evoked release and decreased docked vesicles at the active zone.

In this submission, the authors have performed further experiments on the double mutants. Specifically, they demonstrate that double mutants (cpx-1 unc-11AD) exhibit decreased evoked responses, indicating that the unc-11AD mutants do not affect the fusion promoting aspects of complexin. They describe a more nuanced model, that distinguishes the positive roles of complexin in docking, and inhibitory role of the curvature sensing domain.

I only suggest that they include references to the Snead et al. 2014 manuscript in the Introduction when they state that the C-terminal domain is curvature sensing.

(2) Calcium. A concern was that at 1 mM calcium mini rates were maxed out and could not distinguish differences between unc11AD and the double mutant with complexin. The authors demonstrate that the mini rate in 0.25 mM calcium are identical. This is very convincing data indicating the Acessory Domain is acting in the same pathway as complexin - there is not an additive effect.

(3) Actin. The evidence that the accessory domain of UNC-11 interacted with actin was actually via HIPR-1 as an intermediary. The authors performed an experimental backflip to address this issue. They demonstrated that the UNC-11AD could substitute for the actin-binding domain of HIPR, that UNC-11AD binds actin filaments, and that full-length UNC-11 linked PIP2 to actin.

I had only suggested textual changes to the document to reflect alternative possibilities. The authors went well beyond the recommended amendments in addressing our concerns. The new experimental data are convincing and should assuage future readers' concerns. All of my minor critiques were also satisfactorily addressed, making for a smoother read with less ambiguity in some of the figures.

Reviewer #2: The authors have done an excellent job of responding to all of the concerns, including those of the other reviewers as well as my own. I may have missed it, but the only nagging question I have is why the total loss of unc11 does not increase the frequency of spontaneous release--the SVs are bigger and so should also evade inhibition by complexion. Presumably it is due to the additional defects in protein sorting (VAMP2) but the authors should address this in the text.

---

## [Editor Report · Decision Letter 3]

23 Jan 2026

Dear Jihong,

Thank you for the submission of your revised Research Article "Endocytic protein AP180 Assembly Domain regulates synaptic vesicle size and release in C. elegans" for publication in PLOS Biology. On behalf of my colleagues and the Academic Editor, Cody Smith, I am pleased to say that we can in principle accept your manuscript for publication, provided you address any remaining formatting and reporting issues. These will be detailed in an email you should receive within 2-3 business days from our colleagues in the journal operations team; no action is required from you until then. Please note that we will not be able to formally accept your manuscript and schedule it for publication until you have completed any requested changes. We do apologize for the delays as we worked through the backlog from the recent holidays.

PRESS

Sincerely,

Taylor

Taylor Hart, PhD,

Associate Editor

PLOS Biology

thart@plos.org